# Green Silver and Gold Nanoparticles: Biological Synthesis Approaches and Potentials for Biomedical Applications

**DOI:** 10.3390/molecules26040844

**Published:** 2021-02-05

**Authors:** Andrea Rónavári, Nóra Igaz, Dóra I. Adamecz, Bettina Szerencsés, Csaba Molnar, Zoltán Kónya, Ilona Pfeiffer, Monika Kiricsi

**Affiliations:** 1Department of Applied and Environmental Chemistry, University of Szeged, Rerrich Béla tér 1., H-6720 Szeged, Hungary; ronavari@chem.u-szeged.hu (A.R.); konya@chem.u-szeged.hu (Z.K.); 2Department of Biochemistry and Molecular Biology and Doctoral School of Biology, University of Szeged, Közép fasor 52., H-6726 Szeged, Hungary; noraigaz@gmail.com (N.I.); doraadamecz@gmail.com (D.I.A.); 3Department of Microbiology and Doctoral School of Biology, University of Szeged, Közép fasor 52., H-6726 Szeged, Hungary; betti414@gmail.com (B.S.); pfeiffer@bio.u-szeged.hu (I.P.); 4Broad Institute of MIT and Harvard, Cambridge, 415 Main St, Cambridge, MA 02142, USA; mcsaba@broadinstitute.org; 5MTA-SZTE Reaction Kinetics and Surface Chemistry Research Group, Rerrich Béla tér 1., H-6720 Szeged, Hungary

**Keywords:** green synthesis, silver nanoparticle, gold nanoparticle, nanoparticle characterization, antimicrobial activity, toxicity

## Abstract

The nanomaterial industry generates gigantic quantities of metal-based nanomaterials for various technological and biomedical applications; however, concomitantly, it places a massive burden on the environment by utilizing toxic chemicals for the production process and leaving hazardous waste materials behind. Moreover, the employed, often unpleasant chemicals can affect the biocompatibility of the generated particles and severely restrict their application possibilities. On these grounds, green synthetic approaches have emerged, offering eco-friendly, sustainable, nature-derived alternative production methods, thus attenuating the ecological footprint of the nanomaterial industry. In the last decade, a plethora of biological materials has been tested to probe their suitability for nanomaterial synthesis. Although most of these approaches were successful, a large body of evidence indicates that the green material or entity used for the production would substantially define the physical and chemical properties and as a consequence, the biological activities of the obtained nanomaterials. The present review provides a comprehensive collection of the most recent green methodologies, surveys the major nanoparticle characterization techniques and screens the effects triggered by the obtained nanomaterials in various living systems to give an impression on the biomedical potential of green synthesized silver and gold nanoparticles.

## 1. Introduction

Owing to a number of revolutionary developments in nanobiotechnology, the synthesis methods of various nanomaterials seem uncomplicated and straightforward and enable the construction of literally any type and structured nanoparticle designed and tailored to essentially every possible application let it be in industry, technology or medicine. Metal nanoparticles represent a major class of nanomaterials, where singular physicochemical characteristics yield an ideal platform for the exploitation of such nanomaterials (mainly of silver and gold nanoparticles) for electronics, optics, household items, catalysis, and for various biomedical applications as well [1]. Together with the widespread utilization, the exponentially growing need for nanomaterials and the industrial scale production of these nanomaterials, some concerns have emerged mainly from environment-conscious and eco-sensitive individuals, including numerous researchers [2]. These originate from the fact that nanoparticle production places an enormous burden on the environment, since conventional synthetic approaches often require the administration of toxic chemical entities during the production process, which may cause harmful reactions in the environment and possibly in animal and human health; moreover, such unpleasant chemicals might critically restrict the application possibilities and the biocompatibility of the generated particles [2]. Thus, the pressing demand for metal nanoparticles must be accompanied with eco-friendly, cheap and novel synthesis approaches in order to minimize or completely avoid the administration of dangerous chemicals and at the same time diminish the accumulation of hazardous wastes. Safer production alternatives applying gentle solvents, environment-friendly reducing or stabilizing materials or mild experimental conditions, or even involving the application of biological materials—such as plant extracts or biomolecules of plants, or bacteria, fungi or their lysates—are called green approaches [3,4]. These strategies, although currently in an early phase—thus without substantial and reliable experimental background information or know-how—rapidly gather ground owing to their low environmental footprint, easy methodology and low costs. In the following chapters, we summarize the available primary experimental data on nanoparticles synthesized by means of biological entities, the characterization techniques suggested to describe properly the physicochemical properties of the obtained particles and review the different biological activities exhibited by green synthesized nanomaterials, highlighting the major differences in nanoparticle performance in various biological host systems.

## 2. Synthesis

### 2.1. Synthesis of Silver and Gold Nanoparticles by Microorganisms

Since traditional physical or chemical methods of metal nanoparticle synthesis have obvious limitations and disadvantages, green chemical processes have emerged as a new direction in the chemical industry about two decades ago [5]. Ever since, these biologically inspired green syntheses have attracted considerable attention, offering a promising alternative for maintaining economy while protecting the environment. Over the years, a number of innovative, sustainable synthesis methods have been developed to produce metal nanoparticles using mild experimental conditions (such as ambient pressure, pH and temperature), and a great variety of different non-toxic reducing-capping agents and solvents. In these processes, living organisms, cellular extracts or cell-free growth media of biological agents such as bacteria, fungi, yeasts, viruses, algae or plants are employed as green reaction milieu supplying the reducing as well as the capping agents for nanoparticle formation. These biological entities have been considered as biological “nano-factories” (see Table 1) [6]. Biological synthesis protocols offer a clean, highly tunable, and environmentally benign method for producing nanoparticles with a broad range of sizes, shapes, physical, chemical and biological properties and compositions. The so-formed nanoparticles have a huge advantage over conventionally produced materials: they are more environmentally friendly as compared to the materials covering their surface and are also originally natural, thus biocompatible.

Recently, various microorganisms, mainly bacteria and fungi, have been engaged to produce different metal nanoparticles, such as silver, gold, silver–gold alloy, iron, copper, zinc, palladium and titanium nanomaterials [31,32]. The earliest studies in the research area pointed out that microorganisms have always had a direct or indirect interaction with inorganic materials via geochemical biological processes, originating essentially from the beginning of life; therefore, microorganism-assisted particle synthesis should be regarded as a viable green option and shall be exploited even further. The synthesis of NPs via microbes is a bottom-up approach where nanoparticles are formed as a part of a defense mechanism-based detoxification, as a fundamental survival procedure involving oxidation/reduction of metal ions, generating phosphate, carbonate and sulfide metal forms, or volatilization of metal ions [33]. These processes are carried out by biomolecules, such as various proteins, enzymes, carbohydrates, sugars etc., of the microorganism; however, the exact events of the nanoparticle synthesis have not been fully elucidated yet [34]. The difficulty of identifying the precise mechanism and the active components responsible for the generation of nanoparticles lies in the fact that each kind of microorganism interacts in a different way with a particular metal ion and that the morphology, size and surface properties of the nanoparticles formed are greatly influenced by numerous other factors (mainly environmental conditions such as pH, pressure, temperature), not simply by the biological ingredients of the applied organism [35].

Certain metals, such as silver, are well known for their toxic effects; however, some silver-resistant bacteria can accumulate metals on/in their cell wall. This phenomenon is responsible for the idea of the first pioneering silver nanoparticle synthesis using a silver-resistant bacterium *Pseudomonas stutzeri* [36]. Samadi and co-workers demonstrated similar results obtained via *Proteus mirabilis* bacteria [37]. They also showed that a change in the culturing parameters can massively influence the formation of nanoparticles. Although this conclusion seemed quite unpleasant, it also offered the possibility of modulating nanoparticle features by varying the experimental conditions and shifting the particle syntheses into a more favorable outcome.

Nanoparticles can be generated by microbes either intra- or extracellularly [38]. The intracellular mechanisms involve three main steps called trapping, reduction and stabilization. They rely primarily on the transport of the metal ions into the microbial cell wall. The method involves electrostatic interactions between the negatively charged cell wall and the positively charged metal ions. Then, enzymes residing within the cell wall reduce the toxic metals to harmless nanoparticles, and subsequently, these particles diffuse through the cell wall. Several reports suggested that metal NPs, such as silver and gold, can be easily and readily biosynthesized intracellularly. For example, by using *Pseudomonas* and *Bacillus* strains, small, monodispersed gold nanoparticles were produced [39]. Nair et al. successfully extended this synthesis method to prepare silver, gold and silver-gold alloy nanoparticles [40]. It was also proposed that the formation of nanoparticles using certain yeast strains could carry the greatest potential for nanoparticle manipulation, especially in case of maneuvering nanoparticle shape and size, by controlling culture parameters such as growth and other cellular activities [41].

As for the extracellular nanoparticle synthesis, metal ions on the surface of the cells are converted to metal nanoparticles by microbial enzymes, generally by nitrate reductase or hydroquinone-mediated redox reactions [42]. Successful extracellular biosynthesis of silver and gold nanoparticles was achieved using *Aspergillus*, *Fusarium* and *Rhodopseudomonas* strains [43,44,45]. Moreover, Lengke et al. reported that during extracellular synthesis using the cyanobacterium *Plectonema boryanum*, the size and shape of the formed nanoparticles could be controlled very simply, simply by varying the external temperatures [46]. It is noteworthy that the recovery of metals from the environment by their adsorption onto bacteria also results in bioreduction, yielding metal nanoparticles [47].

In addition to the above described examples, several strains, such as *Pseudomonas fluorescens*, *Geobacillus stearothermophilus* and *Staphylococcus epidermidis* have been successfully applied for the bioproduction of spherical gold nanoparticles in the size range of 5 and 90 nm [48]. Shape selectivity was again observed upon varying the culture conditions.

Gold and silver nanoparticles in various shapes (such as spherical and triangular) have been synthesized using algal strains such as *Padina gymnospora* and *Ecklonia cava*. One such study showed that the astaxanthin-containing green alga *Chlorella vulgaris* can also be applied for gold nanoparticle synthesis [49]. Based on this finding, we utilized *Phaffia rhodozyma* (perfect state *Xanthophyllomyces dendrorhous*), a basidiomycetous red yeast with high astaxanthin content for microbe-assisted nanoparticle synthesis [50]. The cell-free extract of *P. rhodozyma* provided almost monodisperse, well-separated and spherical silver and gold nanoparticles with a narrow size distribution (see Table 1) [5].

Numerous yeasts, fungal and actinomycete strains, and even viruses, have been utilized to assemble gold nanoparticles to form microstructures [48]. Nevertheless, a high number of studies highlighted that among microorganisms, fungi-mediated syntheses hold major advantages over bacteria-, algae- or virus-assisted approaches. They justified this rationale as metal ion conversion to nanoparticles by means of fungal cells offers the easiest and most straightforward procedures to control nanoparticle size, shape and achieve monodispersity. As an example, fungi and yeast strains were used to demonstrate that by varying the pH and temperature during culturing, the size and shape of gold particles can be perfectly adjusted, and that decreasing the pH results in nanoplate formation instead of nanoparticles [45].

Despite the huge potential of using microorganisms for nanomaterial production, there are some limitations which should be considered before use. In fact, using biological agents for the synthesis of silver and gold NPs is preferable over chemical methods due to the simple, eco-friendly approach and also for minimizing the application of harmful chemical solvents and reagents. However, after carefully examining the above-presented microbe-assisted syntheses and the applied biological entities, it can be concluded that despite these approaches being relatively straightforward and favorable, they require specific and rather tedious preparations and multistep processes such as culture isolation, maintenance or growth and inoculum standardization. Moreover, on the surface of the obtained particles, multicomponent residuals from microorganisms can accumulate, which would not only define the physical, chemical and biological characteristics of the obtained nanomaterials and their fate in the presence of living systems, but would also trigger potential immunological reactions after entering the organism. For this reason, the synthesis of metal particles using plants or any part of plants came forth and were later more and more prioritized. These reactions tend to be faster than those performed by microorganisms, are more cost effective and are relatively easy to scale-up for the generation of larger amounts of nanoparticles.

### 2.2. Synthesis of Silver and Gold Nanoparticles by Plants

As discussed above, nanoparticle synthesis using microorganisms is often rather slow, as the availability and maintenance of the various species used in the process is difficult and expensive; moreover, their application on a large scale is fairly restricted [51]. On the other hand, plant-mediated synthesis of metal nanoparticles grants numerous benefits over chemical, physical and microbial methods due to its rapid, well-reproducible, ecological, environmentally friendly, inexpensive procedure that can also be applied readily on an industrial scale [52,53,54]. Therefore, utilization of biological extracts obtained from different plant parts (leaf, fruit, seed, stem, callus, peel and root) for the production of metal nanoparticles such as silver and gold has attracted an extensive amount of interest from the nanobiotechnology research community. As a consequence, a plethora of research papers has been published in the last decade dealing with synthesis approaches using plants, plant extracts or biomolecules deriving directly from plants or plant parts (see Figure 1 and Table 2 and Table 3) [55,56]. Plants contain complex structures that can be used in the reduction and stabilization of the nanoparticles [57]. Plant materials generate nanoparticles by taking up, utilizing, accumulating and using different nutrients [58]. The general protocol for a typical plant-mediated metal nanoparticle synthesis requires first the collection and the purification of the plant part of interest [59]. The plant piece is then dried and powdered. For the plant extract preparation, usually, deionized distilled water is added to the plant powder according to the desired concentration. This solution is boiled and finally filtered. A certain volume of the extract is mixed with the appropriate amount of metal salt solution and the mixture is heated to the necessary temperature for the prescribed time under efficient mixing. To achieve the desired nanoparticles, optimization of every protocol is mandatory using different temperatures, solvents, pH conditions, extract concentrations and incubation times [60,61]. The reduction of metal ions to metal nanoparticles results in a color change of the solution, which can then be monitored by assessing UV-visible spectra. The obtained nanoparticles are usually further characterized using an X-ray diffractometer, scanning or transmission electron microscopy (for the characterization methods, please refer to the next chapter of the present review).

The possible mechanisms of nanoparticle formation using plant extracts have been examined by several authors [63]. Two main theoretical directions have been suggested: 1. Some studies proposed that the bioreduction of the metal ions was the result of trapping these on the protein surface due to electrostatic interactions between the metal ions and the proteins in the plant material extract. Proteins would reduce the metal ions, which ultimately leads to a change in the secondary structure of proteins and also to the formation of metal nanoparticle seeds or, as these are called, nuclei. The formed nuclei increase gradually in size upon accumulation and further reduction of metal ions on the nuclei, leading to the formation of nanoparticles [141]. 2. The second, generally more accepted approach is that the key mechanism behind the plant-mediated synthesis of nanoparticles is a plant-assisted reduction of the metal ions due to various phytochemicals [142]. Based on available literature data, it is probably not one biomolecule that is responsible for the reduction of metal ions, but several plant components and secondary metabolites together are accountable [106,143]. Such active components include various proteins, among them numerous enzymes, amino acids, vitamins, polysaccharides, alkaloids, polyphenols, flavonoids and organic acids, which are known to be non-toxic and biodegradable, and during nanoparticle synthesis, these can act both as reducing and capping agents, thus promoting the formation of nanoparticles and inhibiting their agglomeration [144].

On Au^+^-dihydromyricetin [145] and Ag^+^-hydrolysable tannin [89] pairs, the mechanism of particle formation upon the reduction of metal ions with a plant extract has been analyzed in detail. Metal ions first form complexes with the phenolic hydroxyl groups in the biomolecule; then, the ions are reduced to zero oxidation state metals, while the biomolecule is oxidized, which is then capable of stabilizing the particles in parallel (“capping”). Therefore there is no requirement for the addition of further capping and stabilizing agents during the synthesis. These syntheses are generally very simple: the nanoparticles form spontaneously by mixing the metal salt solution with the plant extract. The formation time of the particles varies between a few minutes and a day, depending on the metal–plant extract pair used. It is also worth noting that the conjugated π-electron system of polyphenols and flavonoids allows the donation of electrons, or hydrogen atoms, from the hydroxyl groups to various free radicals, so these molecules have an antioxidant capacity, which can also extend the life of nanoparticles [146].

The properties and biological performance of silver and gold nanoparticles, which have been generated using plant extracts, are summarized in Table 2 and Table 3 [147]. In 2003, Shankar and colleagues were among the first to report the rapid green production of silver nanoparticles [148]. In their experiments, large amounts of silver nanoparticles were synthesized using plant extracts made from the leaves of Geranium (*Pelargonium graveolens*) and Indian lilac (*Azadirachta indica*) mixed with an aqueous solution of silver nitrate [149]. Among the indisputable advantages of the process, the authors highlighted first of all the speed of the synthesis. The nanoparticles formed much faster during the reduction with the plant extract than in their previous experiments using microorganisms. In the same year, as another step toward plant-mediated nanobiotechnology, Gardea-Torresdey and colleagues reported the production of AgNP particles using Alfalfa (*Medicago sativa*) [150]. Since then, the synthesis of metal NPs has been performed by different research groups utilizing a great variety of plants, plant extracts and their molecular components, where the bioactive materials were used as reducing and stabilizing agents in the production of silver nanoparticles, e.g., extracts of camphor tree (*Cinnamomum camphora*), lemon balm (*Melissa officinalis*), peppers (*Capsicum annuum*), Japanese red pine (*Pinus densiflora*), ginkgo (*Ginkgo biloba*), kobus magnolia (*Magnolia kobus*), oriental planetree (*Platanus orientalis*) and common grape vine (*Vitis vinifera*) [4,151]. These plants contain large amounts of active compounds (e.g., polyphenols, flavonoids) that are suitable for the reduction of metal ions [64,152]. Baharara et al. suggested that phenolic groups and proteins in plant extracts are responsible for the reduction of silver ions [65]. Ajitha et al. showed that the reduction of silver ions can be attributed to the hydroxyl and carbonyl groups in the active components (e.g., flavonoids, terpenoids, phenols, proteins) of plant extracts [66]. Furthermore, they revealed that proteins and peptides form a protective coating around the particles, thereby increasing the stability of the particles and preventing their aggregation. Nadagouda and colleagues were the first to produce silver nanoparticles using coffee and tea extract. In their work, they demonstrated that in addition to the use of plant extracts, no other stabilizing agent was required, as the active ingredients of the extracts served as both reducing and stabilizing agents during the synthesis [153]. This simple one-step synthesis method has also been successfully extended to produce palladium, gold and platinum nanoparticles.

Silver nanoparticles have also been produced with aqueous-alcoholic solutions of roasted coffee (*Coffea arabica*), green tea and black tea [54,58,59]. The authors found that caffeine and theophylline in the extracts were responsible for stabilizing the produced nanoparticles. Moreover, Dhand et al. described that chlorogenic acid is the major phenolic component in the coffee extract that plays an essential role in the reduction of silver ions [64]. Importantly, it has been proposed that the quality as well as the quantity of the potential reducing or stabilizing components of the plant extract used for the synthesis determine the properties of the resulting particles (e.g., size, morphology), including their reactivity in subsequent reactions. Ashokkumar and co-workers observed that particle size decreased with increasing plant extract concentration, while—as described in another study—the number of particles formed correlated with the amount of plant extract used [89]. Moreover, shape selectivity was observed by varying the dose of the bioreducing agent. Chandran et al. also achieved shape and size selectivity of produced silver nanoparticles by modulating the concentration of the starting metal salt solution and aloe vera plant extract [154]. Loo et al. produced round-shaped silver nanoparticles with green tea extract [155]. They made the observation that when the concentration of the extract was increased, the size of the nanoparticles decreased while the number of the particles increased [156].

Recent works pointed out that besides the nature of the plant extracts and the types and concentrations of the active biomolecules within, several factors, including reaction time and temperature, pH and the electrochemical potential, can have an effect on the reduction process [157]. For instance, it was demonstrated that increasing the temperature can improve the nucleation rate, leading to the synthesis of smaller AgNPs and to increased synthesis rate of AgNPs. Furthermore, it was also proven that proteins in the plant extract significantly affect the shape, size and yield of nanoparticles during synthesis [158,159]. Green synthesis of silver nanoparticles was performed using the aqueous solution of *Ziziphus mauritiana* leaves extract as a bio-reducing agent. In this study, the effects of the leaf extract and silver nitrate concentrations, as well as of the temperature on the preparation of nanoparticles were investigated in detail [62].

These green synthesized silver nanoparticles were often produced for specific application purposes—not necessarily for medical utilizations—and sometimes, only the possibility of nanoparticle synthesis using a given plant extract was tested. However, by their application, these nanoparticles could come into contact with living systems; therefore, several research groups following the generation of nanoparticles, rightly, examined their impact on different biological systems [160]. Despite these attempts to assess the effects of the as-prepared nanomaterial on some living organism, only a few studies have examined thoroughly, or compared the complex (antibacterial, antifungal, antiviral and cytotoxic) biological activity of the produced nanoparticles [161]. To follow the most approved characterization procedure, we have investigated the chemical and biological characteristics of nanoparticles prepared by coffee and tea extracts [69]. Our results clearly showed that green materials used for stabilization and for reduction of metal ions have a defining role in the biological activity of the obtained nanomaterial against bacteria, fungi or human cells. Based on our results, we have recommended to obtain a circumspect selection of the green extracts used for the synthesis of nanoparticles, and suggested that a comprehensive screen of the products should be carried out prior to their applications to delineate their behavior in the presence of living systems. Based on today’s nanotechnology results, combining the available methods of biology, chemistry and material sciences, more complex investigations can be carried out [162,163]. Using such an approach, systematic examinations regarding AgNP aggregation behavior with simultaneous measurements of its effect on biological activity can be performed to offer new frontiers to preserve nanoparticle toxicity by enhancing colloidal stability [70].

The synthesis and utilization of gold nanoparticles is an emerging research area due to the unique and tunable surface plasmon resonance and the electrical conductivity, excellent catalytic activity and the biomedical potential of AuNPs, including drug delivery, molecular imaging and biosensing [164]. Therefore, there is also a growing need for environmentally benign synthesis processes of these particles without losing sight of the major aim, i.e., to provide a safe application and avoid adverse effects in medical applications. To date, several examples of gold nanoparticles produced by plant extracts have been reported in the literature (see Table 3) [165,166]. We observed that the same types of plants and their respective components are generally exploited for the synthesis of AuNPs as for AgNPs. The first to report the fabrication of gold nanoparticles using living plants was in 2002 by Gardea-Torresdey and co-workers [150]. They described the formation and growth of AuNPs inside live alfalfa plants. In their study, alfalfa plants were grown in a tetrachloroaurate ion-rich environment. The absorption of gold metal by the plants was proven by transmission electron microscopy and X-ray absorption measurements. *Sesbania drummondii* seedlings were also successfully applied in a similar system [167]. Beattie and Haverkamp revealed that the bioreduction of gold salts to metal occurs in chloroplasts [168].

Although the idea of utilizing living plants is revolutionary; nevertheless, the purification of the intracellularly formed nanoparticles proved to be a difficult task. Therefore, extracellular syntheses of nanoparticles, which utilize the extracts of plants, have gained immediate popularity [169]. As was mentioned before, one of the first studies on the biosynthesis of metallic silver and gold nanoparticles using plant extracts was performed by Shankar et al. who applied geranium leaf extract during the synthesis [148]. These reactions lasted for 2 days and the generated nanoparticles had various morphologies, such as spherical, triangular, icosahedral and spherical. This method was later optimized using other plant extracts (neem leaf), where a shorter reaction time (~2.5 h) was achieved [149]. The rapid green synthesis of monodispersed, spherical gold nanoparticles with dimensions of ~20 nm was observed using *Mangifera indica* leaf extract [170]. The reduction of gold cations to gold nanoparticles by this extract was completed within 2 min and the obtained colloid was found to be stable for more than 5 months. Highly stable crystalline gold NPs were produced using *Momordica charantia* as well [171].

Dwivedi et al. also reported about rapid biosynthesis of metal nanoparticles using *Chenopodium album*, where the leaf extract was successfully applied to obtain silver and gold nanoparticles in the size range of 10–30 nm. They observed not only shape selectivity, but noted that the formation of spherical nanoparticles was more favorable at higher leaf extract concentrations [172]. The synthesis of gold nanoparticles of various shapes (spherical, hexagonal and triangle) via olive leaf extract as a reducing agent has also been demonstrated. The size and the shape of gold NPs were modulated by varying the ratio of plant extract and the initial metal salt in the reaction medium [173]. The authors emphasized the role of the high phenolic content of the hot water extract of olive leaves, which helped in the reduction. They observed that the generated spherical particles were capped by phytochemicals. It is well known that plant and plant-based phytochemicals are rich in various polyphenols, flavonoids, terpenoids, aldehydes, proteins, alkaloids, acids and alcoholic compounds [71,174]. These active components are assumed to participate in the reduction of chloroauric acid to form AuNPs and serve as stabilizing agents to prevent particle aggregation. Smitha et al. achieved shape diversity, when *Cinnamomum zeylanicum* leaf broth was used as a reducing agent: the plant extract at lower concentrations caused formation of prism-shaped particles, while at higher concentrations spherical particles dominated [175]. Tansy fruit extract was successfully employed for the development of silver and gold nanoparticles with spherical and triangular shapes with an average size of ~15 nm [176]. These nanoparticles were found to have a crystalline structure with face-centered cubic geometry verified by the XRD method. Plant extract of *Pulicaria undulata (L.)* was used as both reducing agent and stabilizing ligand for the rapid and green synthesis of gold, silver and gold-silver bimetallic alloy nanoparticles. These nanoparticles showed composition-dependent catalytic activity [177].

Regarding the collected data on metal nanoparticle synthesis, one thing is certain, the possibilities are endless—almost any plant, or part of it, can be put to use for producing nanoparticles; gold NPs were prepared by an environmentally friendly, one-step synthesis process using leaf extracts from coffee, mint, mango, tea, grapes, lemon, eucalyptus, neem, roses, aloe vera, tamarind, coriander and peppermint [178,179,180,181]. However, the careful analysis of the synthesis approach and the physico-chemical properties of the realized nanomaterials accentuate the attentive selection of plant material for nanoparticle synthesis, as these biocomponents are inherently responsible for the morphology, stability and biological properties of the formed NPs.

Apart from the green material, pH can also influence particle production. Armendariz et al. investigated the synthesis of gold NPs using *Avena sativa* at different pH values [182]. It was revealed that at higher pH, small sized NPs were formed, and they suggested that aggregation occurs at lower pH. Their findings imply that variation in the pH of the reaction medium has also a defining role on the shape of the obtained particles [183,184].

Gold nanoparticles obtained by plant-mediated methods have a very particular surface with different structural and functional compounds attached to the metallic surface with varying interactions, which should be considered upon their potential biomedical applications, especially in antimicrobial settings and in human therapy [185]. Plant synthesized and capped gold nanoparticles generally exhibit biocompatibility and have a great antimicrobial and cytotoxic potential. For such applications, it is essential to produce gold NPs with preferentially small sizes but with high biostability.

Compared to conventional methods, plant-mediated syntheses offer an environmentally and economically friendly alternative for the production of metal nanoparticles. Therefore, the exploration of easily accessible plants is recommended, and the development of innovative green synthesis methods that can be implemented on an industrial scale should be encouraged. However, for the safe use of green synthesized metal nanoparticles with different properties in everyday life, a comprehensive study of the nanoparticles is required before their application to assess their behavior in living systems. Acknowledging this concept, in the following chapters, the antimicrobial and anticancer properties of the green produced silver and gold nanoparticles will be summarized and discussed.

### 2.3. Challenges Associated with Green Synthesis

The potential application, fate and behavior of nanoparticles in the environment and towards living systems rely on important functional properties such as size and shape, monodispersity, surface charge, plasmonic response, medical diagnostics response and biofunctional or catalytic activity. Controlled synthesis of the designed greener products by safer processes, while maintaining nanoparticle function and efficiency, is one of the most challenging and recurrent issues to be solved for the development and spread of novel green synthesis protocols [186]. As the organisms used in nanoparticle synthesis can vary from simple prokaryotic bacterial cells to complex eukaryotes, the application of green, sustainable synthesis routes for the production of metal nanoparticles still require extensive research and innovative solutions to set a promising and sustainable trend.

It can be clearly seen from the above-presented synthesis examples that these methods are still in the development stage; therefore, further optimization of these processes is required. The main limiting issue for the large scale and routine utilization of green synthesis is that these nanoparticle systems are very diverse. The experienced problems and the challenges are all associated with the varying type, quality and concentration of green extracts, the diverse ratios of the reagents, reaction conditions (time, temperature and pH) and yield and of the deficient or inadequate product characterization of the obtained particles, which render proper comparisons between nanoparticle performances fairly difficult. Further main issues relate to the control of crystal growth, size and morphology, dispersity and nanoparticle stability [187].

To date, most reports on green synthesis of metal particles using extracts of microorganisms or plants focus mainly on demonstrating the feasibility of these extracts for the nanomaterial synthesis. However, the organic compounds/biomolecules responsible for metal ion reduction are rarely identified, the exact reduction mechanisms are almost never determined and comparisons with conventionally produced nanoparticles are reported only in a few studies [70,188]. Thus, questions arise on whether and how can these methods be adopted for large-scale production of metal nanoparticles to meet industrial needs.

To achieve this goal, future investigations should move toward the optimization of green reaction conditions, to the development of accurate and reliable synthesis protocols that enable the control of the above-mentioned critical factors that affect the properties of nanoparticles [189]. The proper selection of the green reducing agents is of vital importance, as the biological properties of gold or silver nanoparticles produced by microorganisms and plant extracts varies enormously depending on the biological materials used for the synthesis [69]. Plants are generally considered to be ideal candidates for nanoparticle synthesis [156]. However, for a larger scale plant-based nanoparticle synthesis, the green entity should originate from the same site, must be of the same quality, of the same composition in each synthesis round to maintain reproducibility and to guarantee the identical performance of the generated nanoparticles. This requirement is the main limitation of the scale-up manufacturing and of the widespread application of the green synthesized nanoparticles. It is evident that the major challenge is to collect the same material in high quantities, and to grow it in the same way, under the same circumstances to yield identical products. To ascertain replicability, the composition of the biomaterial should be precisely determined prior to be used in nanoparticle synthesis. The type of the green entity and its composition is crucial, as it was shown that nanoparticles of various shapes and sizes were synthesized with differing tea extracts because of the different concentration of caffeine/polyphenols, which acted as both reducing and capping agents upon synthesis [190]. The lack of standard protocols and guidelines for the characterization of the extracts is a serious hurdle in the commercial production of green nanomaterials.

Optimization of the synthesis parameters is also an important aspect which should be considered [144,186]. In order to use microorganisms or plants for synthesis of metal nanoparticles on industrial scale, the yield and the production rate are important issues. Therefore, the concentrations and the ratio of the extract and the salt, the experimental conditions such as synthesis time, pH and temperature, the buffer to be used and the stirring velocity upon production need to be properly controlled and optimized. The size and the shape selectivity, as well as the concentration of the nanoparticles, can be modulated by optimizing the concentration of the green extract [154]. Furthermore, the stability and the aggregation propensity of the obtained nanoparticles can also vary depending on the molecules of the green solution. To precisely design the properties of the nanoparticles, the exact composition (qualitative and quantitative) of the green material, let it be a plant extract or a microbial lysate, should be determined. The more complex the composition of the green solution is, the more problematic the estimation of the expected nanoparticle features becomes. This is generally the biggest problem with green nanoparticle synthesis, since in most cases it is impossible or not worthwhile to determine the components of the green extract. Regarding other experimental factors, it was demonstrated that the longer the reaction time was, the larger the nanoparticle size became [191]. In addition, by increasing the stirring time, the mean particle diameter increased [192]. It was also proven that the mean particle size and the number of nanoparticles formed decreased at higher temperatures [156] The binding of metal ions to the biomolecules of the green material was shown to be pH-dependent; thus, at different pH values, diverse—tetrahedral, hexagonal platelets, rod shaped, irregular shaped—particles can be formed. Generally, smaller nanoparticles were achieved at higher pH [182].

Moreover, extraction and purification of the synthesized nanoparticles from living or non-living biological sources for further applications is still an important and challenging issue. As demonstrated, besides various plants, several microorganisms, including bacteria, fungi and yeasts, are able to produce intra- and extracellularly metal nanoparticles. In case the intracellular approach is applied, the access and the purification of the produced nanoparticles from the cells is extremely complicated. High energy consumption-related physical and/or chemical extraction methods, including heating, freezing and thawing, sonication processes or osmotic shock and several centrifugation and washing steps, should be applied to separate nanoparticles [144,193]. For example, gold nanoparticles were extracted from *Lactobacillus kimchicus* with ultrasonication and continuous centrifugation [194]. Importantly, these procedures are multi-step methods and have large energy and solvent demands, leading to high wastefulness. Furthermore, these methods can affect nanoparticles seriously, by modulating characteristic properties such as shape and size, and to a larger extent the surface properties of nanoparticles [195]. By their utilization nanoparticle aggregation, sedimentation and precipitation could happen, leading to undesired and uncontrolled properties and behavior upon utilization. Another approach for nanoparticle extraction is by enzymatic lysis; however, the particles outside bacteria are generally less stable than those inside the cells and serious aggregation might follow the extraction. Moreover, due to the expected costs, this purification method could not be scaled for industrial production.

The extracellular type of nanoparticle formation is a more favorable method, not only because of the simplicity of purification, but also due to the advantageous production rate [196]. There is no need of downstream purification steps for the extraction, but generally, after the one-step synthesis procedure, centrifugation or filtration is used for purifying nanoparticles [197]. For example, silver nanoparticles synthesized extracellularly by *Pseudomonas sp.* were collected by centrifugation at 12,000× *g* and 25 °C for 10 min and washed three times by water to remove the unconverted metal ions or any other constituents [14].

However, it should be noted that application of these purification steps could also change characteristics such as the stability of the nanoparticles, and as a consequence, particle aggregation is not uncommon. Due to these drawbacks, most studies omit the purification step or apply only a mild filtration or centrifugation step. Nevertheless, green entities not only provide a natural capping to synthesize nanoparticles, but prevent their aggregation by providing additional stability [193]. Considering the scalability potential of the whole nanoparticle formation process, the task of how nanoparticle purification methods affect and change properties of nanoparticles also poses a major challenge.

In addition, there is only a handful of evaluations regarding the complex toxicity and life cycle assessments of nanoparticles as a product. Such assessments should be carried out to estimate the costs and consequences of implementing these synthesis protocols. The main limitation of conducting these quantitative analyses of sustainability implications is the lack of information [198]. There is no experimental data on nanoparticles from all the life cycle stages, and information associated with their synthesis, mechanism, characterization, application, pathway and fate is also scarce. These assessments are not completed due to the following huge limitation: since the reduction mechanism of metal ions to nanoparticles is not yet elucidated, the reducing agent, which is a defining factor for the nanoparticle synthesis, is excluded from most life cycle analysis calculations, which led to constrained modeling and consequent false quantitative assessment of such processes [198]. It also complicates the assessment of these green synthesis methods, in that there are no uniform regulations at either national or international levels that would regulate and describe the quality and the origin of the materials and restrict the conditions of nanoparticle production, standardize and control the quality of nanoparticles formed as well as monitor their use and long-term effects. Therefore, it is a very difficult task to design, estimate and compare the behavior of the nanoparticles with different properties produced by different green entities. Based on the above-described challenges, detailed in-dept research is needed to establish green synthesis methods that are uniform, safe and economic, to determine the effects and potential toxicity of the designed and obtained nanoparticles.

## 3. Characterization

According to the information summarized in the previous chapter, a massive number of biological entities have been successfully applied for metal nanoparticle generation [199]. Although green syntheses are regarded as biocompatible and safe procedures, depending on the extract/microorganism used, materials with different physical, chemical, optical, electrical, biological and catalytic properties are produced [69]. In contrast to chemical methods, where usually only one reducing agent is responsible for the reduction, in green syntheses, a multitude of different molecules react with metal ions, some of these instantly, others in a lagging manner, and form interactions with the nanoparticles upon production [69]. For monitoring the factors affecting the synthesis, the success and the yield of the synthesis, nanoparticle characterization is fundamental. Such a thorough investigation, where not simply the morphology, size and shape, purity, dispersity and solubility, but also the aggregation propensity, stability and surface characteristics are assessed, would give a detailed overview about the nanoparticle features and helps to estimate the behavior and also the degree of how the nanoparticle would affect the various living systems. Unfortunately, only a few articles present a full physical, chemical and biological characterization of the particles before advertising their utilizations for different settings (see Figure 2) [70].

To gain insight into the nanoparticle characteristics, various analytical techniques can be utilized [200,201]. The most frequently applied techniques are the UV–visible (UV-VIS) spectroscopy, scanning and transmission electron microscopy (SEM, TEM), Fourier transformed infrared spectroscopy (FT-IR), powder X-ray diffraction (XRD), energy dispersive spectroscopy (EDX), atomic force microscopy (AFM), dynamic light scattering (DLS), zeta-potential measurement (ZP), thermogravimetric analysis (TGA), inductively coupled plasma mass spectrometry (ICP-MS), Raman spectroscopy and X-ray photoelectron spectroscopy (XPS) [202].

To start with, a less sophisticated but rather useful “method” during nanoparticle synthesis is the visual examination or color change test, meaning that when the reaction mixture changes its color and turns into a brownish (AgNP) or purple (AuNP) color, it is an indication of metal nanoparticle formation [199]. After the synthesis reaction, UV-Vis spectroscopy is generally used to detect the size and stability of the produced nanoparticles. The unique optical properties of metal nanoparticles allow the interaction and resonance with light causing the appearance of a characteristic surface plasmon resonance (SPR) band. A number of studies demonstrated that the absorption SPR peak of metal nanoparticles are between 200–800 nm, and are typically in the range of 400–450 nm for silver and in the range of 500–550 nm in case of gold NPs [201]. UV-VIS spectroscopic analyses are suitable for determining the size of NPs, as these exhibit a red shift in the SPR peak with increasing nanoparticle size and a blue shift for decreasing size. Many researchers use this technique to explore the effect and relationship between factors influencing synthesis such as pH, temperature, reaction time, amount of plant extract and ion concentration. Bélteky et al. published a study about different factors affecting nanoparticle aggregation and its consequences on cytotoxicity and antimicrobial activity based on UV-VIS and DLS measurements [70].

DLS studies are mainly used for ascertaining the particle size distribution and to obtain the average hydrodynamic diameter of the samples [143]. Zeta potential values give information about the stability of the nanoparticles via measuring the electric charge on the particle surface. Sujitha et al. reported that AuNPs show a lower ZP value when lower concentration of the citrus extract was used during the synthesis, whereas higher ZP values were obtained for the produced nanoparticles when higher concentration of the extract was applied.

XRD is a primary analytical tool in gathering information about the crystal structures, phase identification and crystallite size. Most of the studies reported about the formation of face-centered cubic structured silver nanoparticles according to the XRD measurements performed [203]. In some cases, particles with hexagonal and cubic structures were also observed. EDX analysis could also be used to confirm the structure and purity of the synthesized nanoparticles by determining the elemental composition [143].

Microscopy-based measurements, such as SEM, TEM (high resolution, field emission) and AFM, are considered essential tools for obtaining morphology data (e.g., size and shape) from images taken of the nanoparticles [204]. Moreover, AFM provides a 3D image about the particle, and thus, every dimension, even the height of the nanomaterial, can be calculated. These microscopic techniques are also suitable to gather information about the purity, polydispersity and the surface properties of the resulting particles. This is particularly important when green synthesized nanoparticles are characterized, since several papers have shown that the formed particles were embedded or entrapped in a matrix derived from the biological entities used during the green synthesis.

FT-IR measurements provide the means to examine the surface chemistry and to identify surface residues such as functional groups—often hydroxyl and carbonyl moieties—that reside at the surface following particle production. This technique can be used to reveal which biomolecule/s (phenolics, terpenoids, glycosides, peptides, proteins and tannins) can be responsible for the reduction, capping and stabilization of nanoparticles. As a complementary technique, Raman spectroscopy can also be useful in detecting a variety of chemical species that are joined to the surface of nanoparticles during synthesis [200].

Other relevant characterization methods, such as ICP-MS, XPS or TGA could be helpful to describe the surface chemical structure of the particles, to predict precise chemical composition and the thermal stability of the obtained nanomaterials [205].

## 4. Biological Activity

### 4.1. Antimicrobial Activity of Green Synthesized Gold and Silver Nanoparticles

Since the beginning of the 21st century, the treatment of infectious diseases caused by multidrug resistant microorganisms has become one of the main challenges in human therapy, agriculture and in the food industry. Multidrug resistant strains are unresponsive to many antibiotics; therefore, alternative methods are required for their elimination [206]. Management of viral infections is even more problematic, because of resistance development and of the toxicity of the antiviral compounds [207]. Today, the entirety of mankind experiences that virus-caused pandemics have not only a global impact on human healthcare but also influence significantly the world economy [208].

Silver and gold have long been known as potent biocides [209]. Antibiotics in general have a selective target in the microbial cell; however, gold and silver nanoparticles exhibit a fairly broad spectrum activity. AuNPs act via destroying the cell membrane potential, or by inhibiting the binding of tRNA to the small subunit of the ribosome, thereby hindering the protein synthesis. In the presence of AuNPs, the activity of ATP synthase is impeded, which will ultimately lead to cellular ATP depletion [210]. The effect of silver nanoparticles on microbial cells can be categorized in four major areas: AgNPs interact with the cell wall and the plasma membrane, causing structural alterations and eventually the loss of the semi-permeability of the membrane. After entering the cytoplasm, AgNPs induce ROS production and by binding the phosphate group of relevant molecules, it affects major signal transduction processes. The thiol preference of AgNPs leads to strong interactions with amino acids, by further disturbing the protein synthesis [211]. Based on the above-described multitude of cellular targets and mechanisms of action, the development of microbial resistance against AuNPs or AgNPs is rather unlikely; therefore, these metal nanoparticles gain considerable advantage over other types of antimicrobial agents in the long run. Significant developments in nanotechnology made it possible to generate silver and gold nanoparticles comparable in size with certain biomolecules of the living cells. These advancements were soon followed by an ever-growing interest of microbiologists for the production and utilization of these nanoparticles against pathogenic microorganisms, especially against multidrug resistant strains. However, the amplification of microbiological utilization of nanoparticles raised concerns about the induced environmental burden and urged a scientific impetus for eco-friendlier synthetic approaches to be established [36,66,114].

The biological effect—as well as the antimicrobial activity—of AgNPs and AuNPs depends on their size, morphology and concentration [56]. Nanoparticles smaller than 20 nm have a relatively large size/surface ratio; thus, they bind more efficiently to the surface of the microbial cells and penetrate easily across the cell wall and plasma membrane. Although some studies on the dependence of biological action on nanoparticle morphology revealed that triangular shaped particles are more effective than spherical forms [212], there are no data available on the morphology-dependent activity of green synthesized nanoparticles. The reducing and capping agents applied during the green synthesis can also modulate the biological activity of the nanoparticles. In our previous study, we and our collaborators synthesized AgNPs by using sodium citrate and green tea extract [65]. The size and the shape of the as-prepared particles were similar. When we tested the antibacterial and antifungal activity of these particles at the same concentrations, the green tea-synthesized AgNPs proved to be more effective than the chemically synthesized ones. These results suggest that biologically more active nanoparticles can be produced via green synthesis, even if the physical properties of the chemically and biologically synthesized NPs are very similar.

The antimicrobial effects of green synthesized nanoparticles have been studied on the three main groups of microorganisms (viruses, bacteria and fungi), although very rarely on all three groups (see Figure 2 and Figure 3, and Table 1, Table 2 and Table 3) [62,181]. Generally, the antibacterial features are tested following nanoparticle synthesis and characterization. The most severe knowledge gap is on the antiviral propensity of the generated AgNPs or AuNPs, which needs to be filled urgently. The antiviral activity of AgNPs and AuNPs is probably based on their binding to the viral surface proteins preventing their attachment to the cell surface receptors and impeding also the virus entry to the cells. The other possible antiviral mechanism involves the inhibition of the virus particle assembly inside the host cells [213]. The major reason for the lack of studies on the antiviral activity of silver and gold nanoparticles, also of those produced by green methods, is related to the technical difficulties of properly maintaining viruses for the examinations. The few studies published in the field utilized human or animal viruses, which were maintained in tissue culture or using chicken embryos and the antiviral activity of the NPs were detected by MTT assay [116], by hemagglutination tests [17] or by real-time PCR method [130]. Haggag et al. synthesized AgNPs with the aqueous and hexane extract of two plants, namely *Lampranthus coccineus* and *Malephora luthea*, and tested the activity of the particles against HSV-1 (Herpes simplex virus type-1), HAV-10 (New Avian Influenza virus subtype) and CoxB4 (Coxsackie virus B4). AgNPs produced by hexane extract of *L. coccineus* inhibited the action of all the three viruses while AgNPs produced by hexane extract of *M. luthea* inhibited the infection caused only by HAV10 and CoxB4. The antiviral activity was more effective when nanoparticles were applied before the viral infection, suggesting that AgNPs interacted with the coat protein of the viruses, preventing their entrance to the cells [116].

Gold nanoparticles prepared using garlic extract were tested and proven to be effective against measles virus probably for similar reasons as described above in Haggag’s work: via binding to the viral surface proteins, AuNPs are able to inhibit the attachment of the virus particles to their receptors on the surface of the cells [130]. Avilala and Golla demonstrated that *Nocardiopsis alba*-synthesized AgNPs inhibited the action of Newcastle Disease Virus (NDV) [17]. Antiviral activity of the AgNPs formed by curcumin-mediated synthesis was proven against respiratory syncytial virus (RSV). Biologically synthesized AgNPs, for which medicinal plant extracts (*Andrographis paniculata*, *Phyllanthus niruri* and *Tinospora cordifolia*) were used, decreased the infection caused by chikungunya virus. In this study, however, it is not clear that the observed antiviral activity is owing to the reducing agents or to the AgNP itself, since the antiviral properties of these plants were previously known [214]. Unfortunately, none of the research studies examined the antiviral action of any of the generated metal nanoparticles on SARS-CoV-2, or other coronavirus strains.

On the contrary to antiviral features, the antibacterial and/or the antifungal activity of biologically synthesized NPs have been tested by many research groups that are actively working on green nanoparticle synthesis and their application possibilities [34,36,62]. The most commonly used methods for this purpose are either well or disc diffusion assays. In both of these approaches, the microorganism is spread onto the surface or inoculated into the growth medium. In the case of a well diffusion assay, a well is prepared in the medium and the tested nanoparticle solution is loaded into the well, while in the disc diffusion assay, the nanoparticle solution is loaded to a filter paper disc and then placed onto the surface of the inoculated medium [56,61]. The appearance of the inhibition zone around either the well or the filter disc indicates how effective the action of the nanoparticle is. Both of these methods are cheap, quick and can be easily completed; however, the results are difficult to compare and not fully quantitative. Concerning bacteria and yeasts, the establishment of the number of colony forming units (CFU) in the presence of the nanoparticles, compared to a control sample (without NPs), is also a suitable method to describe the efficiency of the particles. This method is more laborious than the previous ones; therefore, it is used more seldom. Nevertheless, it yields a quantitative result. The growth inhibition of filamentous fungi can be studied by inoculating mycelium disc or sclerotia onto the surface of nanoparticle containing and of a control medium. After the incubation time, the diameter of the colony should be measured and compared to the control, or the number of developed sclerotia can be counted, respectively [215]. The microdilution assay is another method suitable for testing antibacterial or antifungal activities, where the growth rate of the microbes is checked in liquid medium in the presence and in the absence of nanoparticles. The turbidity of the samples is detected after the incubation time [216]. The drawback of this method is that the potential intrinsic turbidity of the nanoparticle colloid itself can influence the results.

The antibacterial efficiency of various green synthesized silver nanoparticles has been studied intensively (see Figure 3, Table 1 and Table 2). These studies revealed that it is a powerful agent even against multidrug resistant species, such as *Pseudomonas aeruginosa*, *Staphylococcus aureus* and *Klebsiella pneumoniae*. Interestingly, the susceptibility of Gram-positive and Gram-negative bacteria to silver nanoparticles differs somewhat; Gram-negative bacteria are more susceptible than Gram-positive counterparts. This difference is related to the distinct structure of their cell wall. Gram-negative bacteria have a thin cell wall consisting of a peptidoglycan layer; therefore, AgNPs penetrate more easily across this structure than across the thick cell wall of the Gram-positive species [217].

Another rather valuable property of AgNPs is their biofilm-eradicating capability [218,219]. Biofilm formation is a very important virulence factor of several pathogenic bacteria and yeasts. In the biofilm, the cells are surrounded by an extracellular polymeric substance. This matrix prevents the penetration of the conventional antibiotics; therefore, matrix-embedded cells are resistant to the antibiotic treatment and can be the sources of chronic or recurrent infections. However, AgNPs are capable of diffusing in the matrix and kill the cells in the inner layer of the biofilm [218]; thus, such a biofilm-inhibiting effect of silver nanoparticles is a paramount feature that should be emphasized more, especially in preventive antimicrobial application settings. The absorption and penetration of silver nanoparticles across the biofilm matrix depends on their size as well as on the surface materials, which in case of green synthesized nanoparticles can be quite complex; therefore, during biological synthesis, the appropriate approach and capping agent has to be chosen with care.

The antibacterial and antifungal effect of biologically formed gold nanoparticles is ambiguous. In some relatively extensive studies, no effect could be attributed to AuNPs [220,221]; however, in other research works, AuNPs proved to exhibit potent antibacterial and/or antifungal activity (see Figure 3, Table 1 and Table 3). It can be speculated that as the interaction between the cell wall of the microorganisms and the nanoparticles is governed by electrostatic forces, it was probably the charge of these interacting surfaces that modulated the effect of AuNPs. Caudill and colleagues demonstrated that an abundant quantity of negatively charged teichoic acids in the cell wall of Gram-positive bacteria can interact with positively charged gold nanoparticles [222]. Similar observations were made for Gram-negative bacteria, where the lipopolysaccharide content of the outer membrane provides the negative charge which faces cationic AuNPs [223]. These latter examples further highlight the importance of the nature and the physicochemical features of the capping materials—derived from the green material utilized for the synthesis—which can influence the surface characteristics of the nanoparticles and thereby modulate their activity, e.g., inhibiting their attachment to the bacterial cell wall [68]. Importantly, these nanoparticle features seem to render them highly efficient agents for the elimination of clinically isolated human pathogenic microorganisms as well as of multi-resistant strains [72,119,224].

### 4.2. Toxicity and Anticancer Activity of Green Synthesized AuNPs and AgNPs

One of the most important advantages of green synthesized AgNPs and AuNPs is their potentially enhanced biocompatibility compared to nanoparticles of the same chemical element synthesized in a classical chemical procedure. Similarly to their non-green generated counterparts, these particles could be exploited in the future in cancer therapy; however, their effects on human cells, not only on cancerous ones, needs to be evaluated. As potential therapeutic molecules, they come into contact with numerous cells of the body; thus, it is essential to investigate thoroughly the cytotoxic activity of these nanoparticles.

Several chemically synthesized metallic nanoparticles display anticancer activity *in vitro* and *in vivo* as well. Green synthesis with plants or other organic materials gives the opportunity to prepare nanoparticle solutions carrying biologically active compounds coming from the applied natural extract, which might result in a modulated anticancer activity, rendering the particle more or even less potent, or toxic towards human cancer and non-cancerous cells. In fact, a large number of green synthesized AgNPs and AuNPs were proven to exhibit anticancer activity (see Figure 3, Table 1, Table 2 and Table 3), but their efficiency and the cellular effects were strongly dependent on the natural extract applied during the synthesis procedure. As a first example, AgNPs prepared using *Dendropanax morbifera* leaf extract displayed a marked anticancer activity against A549 lung cancer cells *in vitro* and induced apoptosis; on the other hand, they were not toxic to HaCaT human keratinocytes. Similarly prepared AuNPs were also non-toxic to HaCaT and A549 cells, which advocates the potential application of *Dendropanax morbifera* leaf extract-AuNPs for drug delivery or diagnostic purposes [77].

Herbal medicinal plants are often preferred as biological entities for nanoparticle green synthesis. *Panax ginseng*-mediated AuNPs were not cytotoxic to HaCaT and 3T3-L1 non-cancerous cells. *P. ginseng*-generated AgNPs did not exhibit any significant cytotoxic effects on HaCaT cells; however, they showed rather detrimental effects for 3T3-L1 pre-adipocyte cells [75]. In a similar approach, Pérez et al. revealed that AgNPs mediated by *P. ginseng* were toxic to B16 murine tumor cells, but were comparatively less harmful for human dermal fibroblasts. On the other hand, *P. ginseng* mediated-AuNPs were non-toxic on either human fibroblast or murine cancer cells [75]. Alsalhi et al. investigated the cytotoxic activity of *Pimpinella anisum*-mediated AgNPs and found that toxicity was higher in colon cancer cells (HT115) than in human primary neonatal skin stromal cells (hSSCs) [83].

Despite numerous studies suggesting the biocompatibility of AuNPs, the anticancer activity of some green synthesized AuNPs was confirmed. As an example, AuNPs synthesized with *Trichosanthes kirilowii* caused cell cycle arrest in G_0_/G_1_ phase and induced apoptosis via activating several apoptotic pathways, involving bid, bax/bcl2 and caspases in colon cancer cells [121]. Similar activities were observed on A549 cells and on HepG2 hepatocellular carcinoma cells treated with green synthesized AuNPs prepared by using *Marsdenia tenacissima* plant extract [122] and AuNPs synthesized with *Cordyceps militaris* mushroom extract [11], respectively. AuNPs obtained with *Nerium oleander* extract also decreased the viability of MCF-7 cells. In this case, a significant cancer cell selective activity was observed, since these green AuNPs did not affect the cell viability of primary non-cancerous lymphocytes. As the *N. oleander* plant extract itself caused significantly lower toxicity on both MCF-7 and non-cancerous primary lymphocytes, as compared to the AuNPs, the authors concluded that the natural extract applied during the synthesis of nanoparticles is only partly responsible for the observed biological effects of these AuNPs [123].

AuNPs synthesized by *Gelidium pusillum* induced apoptosis in MDA-MB-231 cancerous cells; in contrast, fewer apoptotic cells were observed in the non-cancerous HEK-293 samples, despite some DNA fragmentation occurring at a nanoparticle concentration of 150 μg/mL [140]. A very similar approach was used for biologically synthesized AuNPs obtained by the dried fruit extract of *Tribulus terrestris*, but in this case, the effects of two AuNPs with different sizes were examined (7 and 55 nm diameters). The study confirmed that cultures of AGS cells treated with larger sized AuNPs contained a lower number of apoptotic cells than cell cultures treated with the smaller AuNPs [124].

The cytotoxic effect of aqueous *Peltophorum pterocarpum* leaf extract-mediated AuNPs were tested on human normal endothelial cells (HUVEC, ECV 304). No inhibition of cell proliferation was measured, confirming the biocompatibility of these AuNPs in an *in vitro* system. This is one of the few studies where the nanoparticle-induced toxicity was investigated in an *in vivo* model as well. Nanoparticle exposure did not induce any adverse clinical signs or weight changes in C57BL6/J mice. No significant changes were detected in the levels of total glucose, urea nitrogen, transaminases and uric acid in the serum compared to the control group of mice except for triglycerides and cholesterol levels in the AuNP treated group. Moreover, no major histopathological differences were observed among the treated and untreated groups [125].

Oftentimes, when a natural extract is utilized for nanoparticle production, AgNPs as well as AuNPs are concomitantly synthesized. It is generally accepted that mostly AgNPs show high anticancer activity compared to AuNPs; however, this feature can depend enormously on the natural extract used for the nanoparticle synthesis and nonetheless on the tested cell lines. A few studies have attempted to compare the biological performance of the green synthesized metal particles and commercially available, chemically generated counterparts. Both AuNPs and AgNPs produced with rhizome of *Anemarrhena asphodeloides* showed anticancer activity on several tested tumor cells, while neither of them affected the viability of normal pre-adipocyte cells. Furthermore, when the effect of green synthesized AgNPs were compared to commercially available AgNPs, the green AgNPs showed higher cytotoxicity on cancer cells than commercial AgNPs. The reason behind this difference was not established, since ROS production upon the various treatments was not consistent [78]. In another study, no differences were observed between the efficacy of green synthesized (*Artemisia turcomanica* leaf extract) and commercial AgNPs using cancer and non-cancerous cells as both AgNPs induced apoptosis in cancer cells to the same extent [79]. A similar comparative study revealed that AgNPs synthesized with walnut green husk extract showed anticancer activity on MCF-7 cells, while it did not cause a cell viability decrease in normal L929 cells; however, commercial AgNPs decreased the viability of both cancerous and non-cancerous cells at the same level [80]. Black tea extract-mediated AgNPs generated analogous results, as they exhibited lower cytotoxicity against normal human primary fibroblasts, and high toxicity towards A2780 ovarian carcinoma cells and HCT-116 colorectal tumor cells. The components of the tea extract solution had no toxic activity on any of the tested cell lines [81]. These results suggest that green metal nanoparticles have the potential to perform better in anticancer tests than chemically synthesized counterparts, and by their utilization, a certain degree of cancer selectivity can also be achieved.

Although the majority of the toxicity studies are based on *in vitro* experiments, which is fairly understandable, there are occasionally publications revealing the impacts of green synthesized nanoparticles on *in vivo* model systems. Yan He et al. performed *in vitro* experiments first to screen the anticancer effects of *Dimocarpus longan*-mediated AgNPs on lung, pancreas and prostate cancer cells. AgNPs showed great inhibitory effects on the H1299 lung cancer, but they were less effective on the VCaP prostate cancer and BxPC-3 pancreas cancer cells. Treatment of H1299 cells with these green AgNPs induced apoptosis with dose-dependent decreases of NF-κB transcriptional activities. This finding is relevant because activated NF-κB is a key regulator of programmed cell death and is associated with lung cancer progression by the transcriptional regulation of responsive genes [225]. The impact of AgNPs on lung cancer cells were linked to a suppression of bcl-2 proteins, resulting in apoptotic cell death [76,226]. Following *in vitro* examinations, the effect of green AgNPs was investigated *in vivo* on mice carrying H1299 lung tumors. AgNPs could inhibit tumor growth in mice, as significant differences in the tumor size between control and AgNP-treated group was detected. These results indicate that green AgNPs could be effective candidates for the treatment of lung cancer *in vivo* as a complementary to classical chemotherapy [76].

In case of *Ficus religiosa*-synthesized AgNPs, cancer cell viability decreased in a time- and AgNP dose-dependent manner *in vitro*. AgNPs synthesized by *F. religiosa* brought about cell death in A549 and Hep2 cells through the induction of apoptosis by increased generation of ROS and decreased levels of antioxidants. Furthermore, both the extrinsic and the mitochondrial apoptotic pathways were activated in AgNP-treated tumor cells. The following *in vivo* studies performed on rats revealed significant increases in the serum levels of AST, ALT, LDH, TNF-α and IL-6 after oral administration of *F. religiosa*-mediated AgNPs and showed accumulation of silver in liver, brain and lungs. However, the levels of these serum parameters reverted back to normal and the complete elimination of AgNPs was also observed by the end of the wash out period [84].

In recent years, hybrid/composite or core-shell bimetallic or trimetallic nanoparticle systems of gold and silver were also developed, which shifted somewhat the focus of investigations. Green synthesized Ag-Au composites give the opportunity to exploit the anticancer effects of both AgNPs and AuNPs and fine-tune the biological activities of the obtained nanomaterials. In one of these studies, the anticancer effects of starch-mediated bimetallic Ag-AuNPs were investigated. Ag-AuNPs were not toxic to human dermal fibroblasts, while they significantly decreased the viability of melanoma cells. In comparison, monometallic AuNPs synthesized with starch were not toxic to either fibroblasts or melanoma cells [126].

The molecular mechanisms behind the anticancer effect of Ag-AuNP composites synthesized with *Trapa natans* peel extract were investigated on p53 wild type and p53 knockout cells. Ag-AuNPs induced oxidative stress in cancer cells, and caused apoptotic cell death in a ROS-mediated p53-independent way via mitochondrial damage and through activation of caspase-3. It was emphasized that ROS production must be an important factor in the mechanism of action of this green Ag-AuNP composite, because apoptosis was attenuated upon decreasing ROS levels [127].

Obviously, besides plant-mediated nanoparticle synthesis methods, there are other green, cost-effective and rapid techniques utilizing fungi and bacteria for this purpose [227,228]. In cases such as bacteria-generated metal nanoparticles, the thorough biological screening of the potential detrimental effects on living systems is mandatory. AgNPs synthesized by *Escherichia fergusonii* exhibit a toxic effect on MCF-7 breast cancer cells by inducing ROS generation, leading to cellular apoptosis. This study indicated that these bacteria-mediated nanoparticles could have antiproliferative effects as well [30]. Interestingly, human cervical cancer cells were highly sensitive to *Pseudomonas aeruginosa*-synthesized AgNPs and their proliferation capacity decreased with increasing dose of AgNPs [7]. Another *Pseudomonas* species was also proven to be an adequate candidate for nanoparticle synthesis. Gopinath et al. investigated the cytotoxic effects of *P. putida*-generated AgNPs on HEp-2 cells, which revealed that AgNPs did not affect significantly the viability of these cells. It is important to mention that the AgNP concentration used on human cells was lethal for the bacteria tested in parallel. These results indicate that the biogenic, *P. putida*-synthesized AgNPs are capable of displaying antibacterial activity without being harmful to tumor cells at this concentration [8]. The work of Senthil et al. led to a similar conclusion, in that the produced green (in this case plant-based, fenugreek leaves’ extract) AgNPs were less detrimental to human HaCaT cells than to bacteria [82]. The cellular effects of *Bacillus funiculus*-mediated AgNPs were also investigated by assessing cell viability, metabolic activity and oxidative stress on MDA-MB-231 breast cancer cells. Dose-dependent cytotoxicity, activation of caspase-3 and generation of ROS were demonstrated for these AgNPs against MDA-MB-231 cells. The resulting apoptosis was further confirmed by detecting nuclear fragmentation [9]. AuNPs synthesized by *Paracoccus haeundaensis* BC74171^T^ bacteria were biologically inert on normal cells and showed slight toxicity on cancerous cells in the highest applied concentrations [14].

Cyanobacteria, such as *Oscillatoria limnetica*, were also utilized to produce metal nanoparticles, and the cytotoxic potential of the obtained nanomaterials were tested against human cancer cell lines. In case of HCT-116 cells, cyanobacterium-synthesized AgNPs decreased the cell viability more than in the case of MCF-7 cells via inducing apoptosis, which was represented by the morphological changes in tumor cells [10]. Since it is vital to ensure the biosafety of the metal nanoparticles before their actual utilization, the authors quite rightly examined the hemolytic potential of the obtained nanoparticles on human erythrocytes. It was demonstrated that increasing AgNP concentration could induce red blood cell lysis; however, the mode of action for inducing hemolysis was not revealed [10].

The application of fungi as reducing and stabilizing agents in the biologically synthesized AgNPs is engaging due to the production of large quantities of proteins, high yields, easy handling and low toxicity of the residues [229]. *In vitro* AgNPs synthesized by *Agaricus bisporus* showed dose-dependent toxicity on MCF-7 human breast cancer cells. In *in vivo* experiments, it was also demonstrated that the combination of these AgNPs and gamma radiation could induce apoptosis in Ehrlich solid tumor cells in mice via a mechanism involving caspase-3 activation. In Ehrlich solid tumor cells, this treatment combination mitigated significant superoxide dismutase and catalase activities and reduced glutathione levels, whereas it increased malondialdehyde and nitric oxide levels [12]. In another study, treatment of MDA-MB-232 breast cancer cells with *Ganoderma neo-japonicum*-mediated AgNPs reduced the cell viability and induced membrane leakage in a dose-dependent manner. Tumor cells exposed to these nanoparticles showed increased amounts of ROS and triggered hydroxyl radical production. In fact, the apoptotic effects of AgNPs were confirmed by activation of caspase-3 and DNA nuclear fragmentation [13].

### 4.3. Further Biomedical Applications of Green Synthesized AuNPs and AgNPs

In the last few decades, research on efficient metal nanoparticle-based medical approaches for drug-delivery, regenerative medicine, imaging and biosensing have gathered impetus. Provided by this stimulus, alternative, mainly green synthetic methods for silver and gold nanoparticles destined intentionally for such specific purposes have received scientific attention.

Ever since, several articles describing the carrier function of biological synthesized nanoparticles—especially gold nanoparticles—have been published. In one study, protein-coated AuNPs synthesized by *Tricholoma crassum* were found to be promising candidates for gene delivery, since green fluorescent protein (GFP) was successfully carried into mouse sarcoma cancer cells using a plasmid DNA-AuNP complex. Moreover, these AuNPs showed low hemolytic activity towards human erythrocytes, which confirms their biocompatible nature. These results indicated that green nanoparticles could be considered as potential drug delivery platforms of cancer therapeutics [230]. Mukherjee et al. also demonstrated, using *in vitro* systems and an *in vivo* mouse model, that *Peltophorum pterocarpum*-synthesized AuNPs are suitable and effective anticancer drug-carriers. They designed a biosynthesized AuNP-based drug delivery system, in which these particles were conjugated with Doxorubicin (Dox). *In vitro* Dox conjugated-AuNPs showed high antiproliferative activity against A549 lung cancer and B16F10 melanoma cells. They obtained similar results *in vivo*, as a significant reduction in tumor growth was observed compared to the untreated and the unconjugated-AuNP treated group. Significant amounts of conjugated AuNPs accumulated in the spleen 2 h after the treatment; however, 24 h post-injection, the tumors showed a strong tendency to accumulate high levels of Dox conjugated-AuNP. The biodistribution of the drug conjugated-AuNPs reflected the selectivity of this drug-carrier system [125]. Similarly, Patra et al. also investigated a Dox conjugated-nanoparticle-based drug delivery system. They utilized *Butea monosperma* synthesized AuNPs and AgNPs. *In vitro* the drug conjugated-AuNPs and AgNPs showed significant cell proliferation inhibitory activity towards B16F10 cells in a dose-dependent manner compared to free Doxorubicin applied at the same concentration. The enhanced anticancer effects of Dox conjugated-nanoparticles were verified by apoptosis detection [231]. Gellan gum, secreted by bacteria, was used as a reducing agent in the biosynthesis of AuNPs, followed by nanoparticle conjugation with Doxorubicin. The effects of drug-loaded AuNPs were determined on two glioma cell lines. The cytotoxicity of free Doxorubicin and Dox-conjugated AuNPs gradually increased with increasing concentrations; however, the toxic nature of Dox-loaded AuNPs was more prominent and exceeded the same features of free Doxorubicin, indicating the strong carrier potential of Dox-loaded AuNPs. Microscopic experiments demonstrated significant morphology changes and apoptotic cell death triggered by Dox-conjugated AuNPs on LN-18 and LN-229 human glioma cell lines [232].

The toxic effect of resveratrol conjugated-biosynthesized AuNPs (generated by *Delftia sp.* strain) was tested on A549 human lung cancer and on MCR-5 normal fibroblast cells. Resveratrol conjugated-AuNPs showed significantly higher cytotoxic activity towards A549 cells than resveratrol alone. However, no cytotoxicity was observed on non-cancerous MRC-5 fibroblasts [233].

*Punica granatum* synthesized AuNPs were conjugated with the chemotherapeutic agent 5-Fluorouracil (5-Fu). The problem in using this drug is its toxicity to bone marrow and to the gastrointestinal tract. To tackle this problem, Ganeshkumar et al. have developed a method to biosynthesize AuNPs functionalized with folic acid (FA) for targeted 5-Fu delivery. The rationale to functionalize nanoparticles with FA is that folic acid receptors on the cell membrane can be targeted for tumor selective drug delivery. Several liver and breast cancer cell lines are known to overexpress folate receptors; thus, the *in vitro* cytotoxicity of this drug delivery system was investigated on MCF-7 breast cancer cells. Higher cytotoxic effect was measured in case of 5-Fu@nanogold-FA treatment compared to 5-Fu alone or 5-Fu@AuNPs in the same concentrations. The authors found that these drug-conjugated AuNPs functionalized with FA could induce the expression of both p53 and p21 in a concentration-dependent manner in MCF-7 cells. These findings suggest that the JNK/ERK signaling pathway could be involved in p21WAF1-mediated G1-phase cell cycle arrest and growth inhibition in 5-Fu@nanogold-FA treated breast cancer cells [234]. A further study of the same authors dealt with a similar targeted drug delivery system, in which they synthesized pullulan stabilized AuNPs which were coupled with 5-Fu and folic acid (FA) again. *In vitro* cytotoxicity assays on HepG2 hepatocarcinoma cells revealed that 5-Fu@AuNPs-FA exhibit higher toxic activity than 5-Fu alone or 5-Fu@AuNPs, which again pointed to a conclusion that 5-Fu@AuNPs-FA could be a promising alternative carrier for targeting liver cancer [235]. Yallappa et al. utilized AuNPs synthesized by *Mappia foetida* to examine their applicability in targeted cancer therapy. They described that Doxorubicin-loaded AuNPs conjugated with n-hydroxysuccinamide (NHS) activated folic acid (FA) showed low toxicity against Vero normal epithelial cells and high cytotoxic activity against human cancer cells (MDA-MB-231, HeLa, SiHa, Hep-G2) [236].

For diagnostic purposes, biocompatible AuNPs have to be designed and tested. Albeit green synthesized nanoparticles could hold great potential in diagnostics, their biocompatibility is strongly dependent on the natural extract applied upon synthesis. Green magnetite-gold nanohybrids (Fe(3)O(4)/Au) produced with grape seed proanthocyanidin can be suitable for MRI and CT imaging as contrast agents. The magnetite part gives superparamagnetism in MRI, while the gold part of the hybrid provides high X-ray contrast in CT. The nanocomposites are biocompatible and suitable for labeling and imaging stem cells, since nanocomposites could be internalized and accumulated in the cytoplasm of these cells [237]. Cinnamon-generated AuNPs were also shown to be suitable for *in vitro* and *in vivo* imaging. These nanoparticles are not only biocompatible, but their colloid solution is pure enough for *in vivo* applications. Cinnamon-AuNPs are capable of labeling cancer cells *in vitro*, and can be detected by photoacoustic methods; furthermore, with the help of these green synthesized AuNPs, circulating tumor cells can potentially be detected *in vivo* as well. Moreover, biodistribution studies revealed that cinnamon-AuNPs are mostly accumulated in the lungs, indicating their use as contrast agents targeting the lung [238]. Fluorescently labeled AuNPs prepared with *Olax scandens* leaf extract were aimed for both therapeutic and diagnostic purposes. The phytochemicals of *Olax scandens* leaf yield anti-cancer properties to the as-prepared AuNPs, and owing to fluorescent proteins provided by the green extract, these fluorescent nanoparticles enable the detection of the cancer cells [120].

Another application purpose of green synthesized AgNPs could be regenerative medicine. Nanoparticles are mostly applied in such studies because of their wound healing-inducing activity. *Sanghuangporus sanghuang* polysaccharide synthesized AgNPs with chitosan, forming a porous sponge structured matrix, increased wound healing via inducing wound contraction and internal tissue regeneration in damaged skin of animals and disinfected the skin surface to inhibit the growth of *Escherichia coli* and *Staphylococcus aureus* [239]. Green AuNPs synthesized with *Coleus forskohlii* root extract could enhance the wound closure, suppress the inflammation and induce the re-epithelization of excision in Wistar rats [240].

Green synthesized metal nanoparticles are potential biosensor candidates. The shift of the surface plasmon resonance (SPR) peak in the spectra of green synthesized nanoparticles is usually followed, which can also vary depending on the natural extract applied upon synthesis [241,242]. For example, AgNPs synthesized with neem extract exhibit high SPR, while those obtained using guava, mint or aloe leaf extracts resulted in a lower SPR peak. A study reported the capacity of green synthesized silver nanoparticles to detect harmful molecules based on SPR changes, such as different MCZ pesticides in water samples. It was found that MCZ pesticides interact with AgNPs and after UV-visible illumination of the samples, MCZ pesticides can even be damaged and aggregated via the photocatalytic actions of AgNPs [243]. Gold and silver nanoparticles are sensitive materials for the detection of pollutants and heavy metals in environmental samples as well [244,245]. As an example, AgNP colloid synthesized with onion extract was used in a highly sensitive and rapid colorimetric assay for mercury ion detection based on the localized surface plasmon resonance. [246]. Detection of mercury ions at wide pH ranges was also possible by green AgNPs synthesized with *Citrus lemon* fruit extract, supporting the applicability of green synthesized noble metal nanoparticles as biosensors [247].

## 5. Concluding Remarks

Based on the numerous experimental data accumulated in the literature and summarized here in this review, it is evident that the field of green synthesized metal nanoparticles is expanding continuously, and every new prospective emerging on the horizon offers the possibility of finding other, more innovative ways and means to produce silver or gold nanoparticles with the exact properties needed for a specific purpose. Since the biological entities potentially applicable for green synthesis are practically endless, research has to continue to prepare, test and experiment with nanoparticles—synthesized in an eco-friendly approach, using green and renewable materials directly from nature—which exhibit unique properties and behave in the desired manner upon encountering living systems, such as human or fungal cells, bacteria or even viruses. Nevertheless, careful considerations in the selection of the green material for nanomaterial production, and more importantly, a comprehensive screening protocol of these green particles, are obligatory to predict the attitude and performance of NPs on living cells. First of all, the chemical composition of the applicable green material should be considered to estimate which biomolecules have the capacity to act as reducing or capping agents and which of them can be potentially adsorbed on the nanoparticle surface, creating a bioactive coating to interact with living cells upon action. The examples itemized above in this review clearly show that the green materials employed for the synthesis will define or at least fine-tune the chemical and physical properties and surface chemistry and thereby the biological activity of the obtained nanoparticles. After the green material is selected, and its chemical composition and its active ingredients have been regarded, all other chemicals required for nanoparticle synthesis should be attentively picked to preferentially utilize biocompatible substances and to avoid toxic chemicals, leaving only nonirritating, innocuous waste materials behind. When the nanomaterial is readily obtained, a meticulous examination of its structure and physicochemical properties has to be completed to reveal the average size, morphology, surface chemistry and other critical factors. This step is just as important as the synthesis approach, since these findings either promote the nanomaterials for biological tests or advise further optimization of the preparation protocol in case nanoparticles with undesired properties are formed. Finally, a comprehensive biological screening has to be carried out by inspecting the toxicity of the green nanoparticles on various human cell types, on Gram-negative and -positive bacteria, on a number of fungal strains; eventually, the antiviral propensity can be assessed as well. Depending on the original purpose of nanoparticle synthesis, each of these biological characterizations should be broadened by further implementing cell types and strains or even by *in vivo* studies and extending the technical repertoire with additional assays. We cannot stress enough the relevance of performing the outlined characterization route; otherwise, the chemical and biological profile of the obtained green nanomaterial may not be confidently trusted and adverse effects will be observed upon its application.

## Figures and Tables

**Figure 1 molecules-26-00844-f001:**
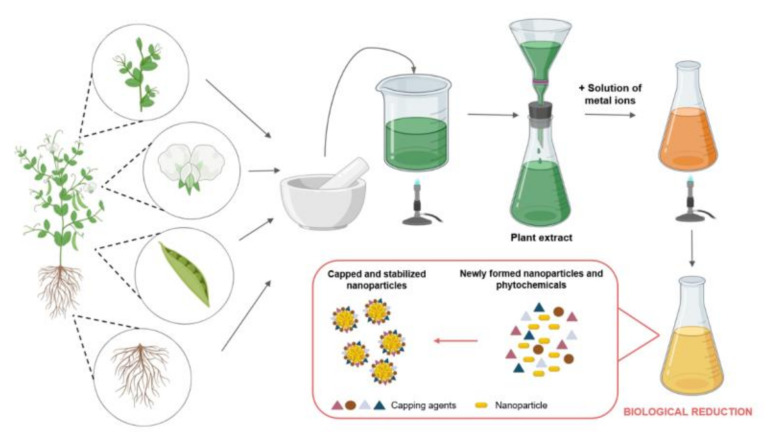
The general steps of green synthesis of inorganic nanoparticles using plant extracts. The figure was created with BioRender.com.

**Figure 2 molecules-26-00844-f002:**
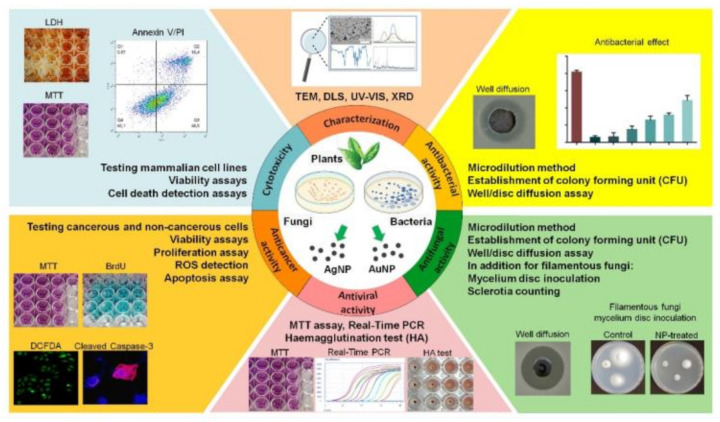
Methodology for the characterization and biological screening of green synthesized silver and gold nanoparticles. The figure was created with BioRender.com. Abbreviations: ROS: reactive oxygen species, TEM: transmission electron microscopy, DLS: dynamic light scattering, UV-Vis: ultraviolet and visible light spectrophotometer, XRD: X-ray diffraction, NP: nanoparticles, LDH: lactate dehidrogenase, MTT: 3-(4,5-dimethylthiazol-2-yl)-2,5-diphenyltetrazolium bromide, PI: propidium iodide, BrdU: 5-bromo-2’-deoxyuridine, DCFDA: 2′,7′-dichlorodihydrofluorescein diacetate, PCR: polymerase chain reaction.

**Figure 3 molecules-26-00844-f003:**
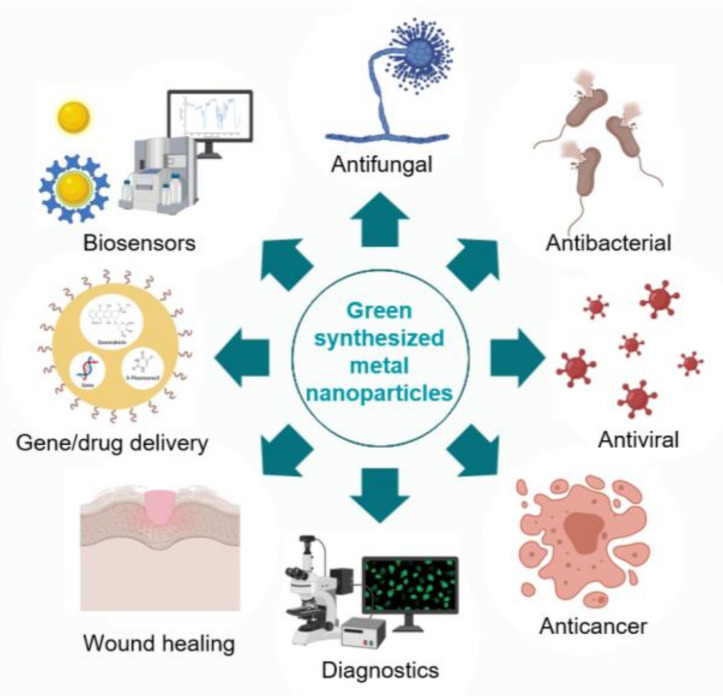
The various biological effects and potential applications of green synthesized metal nanoparticles. The figure was created with BioRender.com.

**Table 1 molecules-26-00844-t001:** Characteristics and biological activities of microbe-mediated silver and gold nanoparticles.

Organism Used for Green Synthesis	Particle Size/Shape	Antimicrobial Effect/ Sensitive Species	Effect on Mammalian Cell Lines	Ref.
*Pseudomonas aeruginosa* supernatant	Ag: 13–76 nm spherical	*Escherichia coli, Vibrio cholerae, Aeromonas* sp., *Corynebacterium* sp., Anti-biofilm activity: *Pseudomonas aeruginosa* and *Staphylococcus aureus*	Cytotoxic (human cervical cancer cells)	[7]
*Pseudomonas putida* supernatant	Ag: 6–16 nm monodispersed, spherical	*Pseudomonas aeruginosa, Escherichia coli, Bacillus cereus, Helicobacter pylori, Staphylococcus aureus*	Non-cytotoxic under 25 μg/mL and cytotoxic at above 50 μg/mL (HEp-2)	[8]
*Bacillus funiculus* supernatant	Ag: 20 nm spherical	n.a.	Cytotoxic (MDA-MB-231)	[9]
*Oscillatoria limnetica* fresh biomass	Ag: ∼3–18 nm quasi-spherical	*Escherichia coli, Bacillus cereus*	Cytotoxic (HCT-16) Less cytotoxic MCF-7) In high concentration hemolytic activity (human erythrocytes)	[10]
*Cordyceps militaris* cell filtrate	Au: 15–20 nm face-center-cubic structure	n.a.	Cytotoxic (HepG2)	[11]
*Agaricus bisporus* filtrate	Ag: 8–20 nm spherical	n.a.	Cytotoxic *in vitro* (MCF-7) and *in vivo* combined with gamma radiation (Ehrlich solid tumor cells in mice)	[12]
*Ganoderma neo-japonicum* mycelia extract	Ag: 2–20 nm spherical	n.a.	Cytotoxic (MDA-MB-231)	[13]
*Paracoccus haeundaensis BC74171^T^* supernatant	Au: ∼20 nm, spherical	n.a.	Slightly cytotoxic (A549, AGS) Non-cytotoxic (HaCaT, HEK293)	[14]
*Micrococcus yunnanensis* supernatant	Au: 53.8 nm spherical	*Staphylococcus aureus, Bacillus subtilis, Micrococcus luteus, Salmonella typhi, Escherichia coli, Klebsiella pneumoniae, Pseudomonas aeruginosa*	Anticancer (U87, HT1080, PC12, Caco-2, MCF7, A549) Slightly cytotoxic (NIH-3T3 and Vero)	[15]
*Brevibacillus formosus* supernatant	Au: 5–12 nm, spherical	*Escherichia coli, Staphylococcus aureus*	n.a.	[16]
*Nocardiopsis alba* supernatant	Au: 32.5 nm polydispersed, spherical	*Pseudomonas aeruginosa, Klebsiella pneumoniae, Staphylococcus aureus, Escherichia coli,* New castle viral disease virus (NDV)	n.a.	[17]
*Nocardiopsis* sp. supernatant	Au: 11.57 nm spherical	*Bacillus subtilis, Pseudomonas aeruginosa, Escherichia coli, Staphylococcus aureus, Candida albicans, Aspergillus niger, Aspergillus fumigatus, Aspergillus brasiliensis*	Cytotoxic (HeLa)	[18]
*Bacillus subtilis* cell-free extract	Ag: 10–20 nm, cubic	*Bacillus cereus, Escherichia coli, Klebsiella pneumoniae, Pseudomonas aeruginosa, Salmonella typhi, Staphylococcus aureus, Streptococcus pyogenes, Streptococcus mutans, Aspergillus niger, Candida albicans, Candida parapsilosis, Candida tropicalis, Cryptococcus neoformans*	n.a.	[19]
*Acinetobacter calcoaceticus* cell-free extract	Ag: 8–12 nm, spherical	*Acinetobacter baumannii*	n.a.	[20]
*Pseudomonas hibiscicola* supernatant	Ag: 10–70 nm, crystalline	*Enterococcus faecalis*, *Klebsiella pneumoniae*, *Mycobacterium tuberculosis*, *Pseudomonas aeruginosa*, *Staphylococcus aureus*	Cytotoxic (Vero)	[21]
*Bacillus cereus* supernatant	Ag: 24–46 nm, spherical	*Escherichia coli*, *Klebsiella pneumoniae*, *Pseudomonas aeruginosa*, *Staphylococcus aureus*	n.a.	[22]
*Pseudomonas aeruginosa* supernatant	Ag: 25–45 nm spherical, pseudospherical	*Acinetobacter baumannii*, *Escherichia coli*, *Enterococcus faecalis*, *Klebsiella pneumoniae*, *Staphylococcus aureus*, *Staphylococcus epidermidis*	Less cytotoxic (human neutrophils)	[23]
*Xanthomonas* spp. fermentative medium	Ag: ˂10 nm, spherical	*Acinetobacter baumannii*, *Pseudomonas aeruginosa*	n.a.	[24]
*Escherichia hermannii* supernatant	Ag: 4–12 nm, spherical	*Escherichia coli*, *Klebsiella pneumoniae*, *Staphylococcus aureus*, *Staphylococcus epidermidis*, *Pseudomonas aeruginosa*	n.a.	[25]
*Citrobacter sedlakii* supernatant	Ag: 4–15 nm, spherical			
*Pseudomonas putida* supernatant	Ag: 4–30 nm, spherical			
*Bacillus endophyticus* biomass and supernatant	Ag: 5.1 nm, spherical	*Escherichia coli*, *Salmonella typhi*, *Staphylococcus aureus, Candida albicans*	n.a.	[26]
*Bacillus brevis* cell filtrate	Ag: 41–68 nm, spherical	*Salmonella typhi*, *Staphylococcus aureus*	n.a.	[27]
*Penicillium polonicum* cell filtrate	Ag: 10–15 nm spherical/near spherical	*Acinetobacter baumannii*	n.a.	[28]
*Phanerochaete chrysosporium* cell filtrate	Ag: 34–90 nm, spherical, oval	*Klebsiella pneumoniae*, *Pseudomonas aeruginosa*, *Staphylococcus aureus*, *Staphylococcus epidermidis*	Minimally cytotoxic (mouse embryo fibroblasts)	[29]
*Escherichia fergusonii* supernatant	Ag: 50 nm mostly spherical	n.a.	Cytotoxic (MCF-7)	[30]

Non-cancerous cells: HaCaT (human keratinocyte cell line), 3T3-L1 (murine pre-adipocytes), HDF (human dermal fibroblasts), human lymphocytes, HUVEC (human endothelial cell line), ECV304 (human endothelial cell line), PBMC (human peripheral blood mononuclear cells), hSSCs (human neonatal skin stromal cells), L929 (murine fibroblast cell line), primary human fibroblast, NIH-3T3 (murine fibroblast cell line), rat splenocytes, HEK-293 (human embryonic kidney). Cancerous cells: B16 (murine melanoma cell line), HCT-116 (colon cancer cell line), A549 (human lung adenocarcinoma cancer cell line), MCF-7 (human breast adenocarcinoma cell line), AGS (human adenocarcinoma cell line), H1299 (human non-small lung carcinoma cancer cell line), VCaP (human prostate cancer cell line), BxPC-3 (human pancreas cancer cell line) MDA-MB-231 (human breast adenocarcinoma cell line), HCT116 (human colorectal carcinoma cell line), SH-SY5Y (human neuroblastoma cell line), COLO205 (human colon adenocarcinoma), HeLa (human cervical adenocarcinoma cell line), Hep2 (human carcinoma cell line), A2780 (ovarian carcinoma cancer cell line), HT115 (colon cancer cell line), HT29 (human colorectal adenocarcinoma cell line), melanoma cells, EAC (Ehrlich Ascites Carcinoma), Jurkat cells (immortalized line of human T lymphocytes).

**Table 2 molecules-26-00844-t002:** Characteristics and biological activities of plant-mediated silver nanoparticles.

Organism Used for the Synthesis	Particle Size/Shape	Antimicrobial Effect/Sensitive Species	Effect on Mammalian Cell Lines	Ref.
*Ziziphus mauritiana* leaf	3–20 nm, spherical	*Escherichia coli*, *Staphylococcus aureus*, *Pseudomonas aeruginosa*, *Bacillus subtilis*	n.a.	[62]
Sugarcane leaf (*Saccharum officinarum*)	20–50 nm, spherical	*Phytophthora capsici*, *Colletotrichum acutatum, Cladosporium fulvum*	n.a.	[61]
*Salvia spinosa* plant	19–125 nm, rounded	*Bacillus subtilis*, *Bacillus vallismortis, Escherichia coli*	n.a.	[63]
*Coffea arabica* seed	10–40 to 20–150 nm	*Escherichia coli, Staphylococcus aureus*	n.a.	[64]
*Achillea biebersteinii* flower	10–40 nm, spherical and pentagonal	n.a.	Cytotoxic (MCF-7)	[65]
*Lantana camara* leaf	14–27 nm, spherical	*Escherichia coli*, *Pseudomonas* spp., *Bacillus* spp., *Staphylococcus* spp.	n.a.	[66]
Black tea leaf	9–15 nm, spherical	n.a.	Cytotoxic (MCF-7)	[67]
Green tea leaf (Richun Tea)	20–90 nm, spherical	*Escherichia coli*	n.a.	[68]
Green tea leaf (R. Twining), coffee seed (Tchibo Family)	3–12 nm, spherical	*Bacillus cereus* var. *mycoides*, *Micrococcus luteus*, *Escherichia coli*, *Pseudomonas aeruginosa, Saccharomyces cerevisiae*, *Candida parapsilosis*, *Candida albicans*, *Cryptococcus neoformans*	Anti-proliferative, cytotoxic (HeLa and NIH/3T3)	[69]
Green tea leaf (R. Twining)	5–15 nm, spherical, polyhedron	*Escherichia coli, Bacillus megaterium,* *Cryptococcus neoformans*	Cytotoxic (A549 and NIH/3T3)	[70]
*Azadirachta indica* leaf	~34 nm, spherical	*Escherichia coli, Staphylococcus aureus*	n.a.	[71]
*Mussaenda glabrata* leaf	55 nm, spherical and triangular	*Bacillus pumilus*, *Staphylococcus aureus*, *Pseudomonas aeruginosa*, *Escherichia coli*, *Aspergillus niger, Penicillium chrysogenum*	n.a.	[72]
*Plumbago zeylanica* bark	10–25 nm, spherical	*Escherichia coli*, *Pseudomonas aeruginosa, Bacillus subtilis, Staphylococcus aureus*, *Candida tropicalis*	Cytotoxic (Dalton Lymphoma Ascites)	[73]
*Panax ginseng* fresh leaf	5–15 nm, spherical	n.a.	Non-cytotoxic (HaCaT 20 μg/mL) Cytotoxic (3T3-L1 20 μg/mL)	[74]
*Panax ginseng* berry	10–20 nm, spherical	n.a.	Cytotoxic (B16) Less cytotoxic (HDF)	[75]
*Dimocarpus longan* dried peel	8–22 nm, spherical	n.a.	Cytotoxic *in vitro* and *in vivo* (H1299) Less cytotoxic *in vitro* (VCaP, BxPC-3)	[76]
*Dendropanax morbifera* leaf	100–200 nm; polygonal and hexagonal	n.a.	Cytotoxic (A549) Cytotoxic in high concentration (HaCaT)	[77]
*Anemarrhena asphodeloides* rhizome	20 nm, crystalline face-centered cubic	n.a.	Cytotoxic (A549, HT29, MCF-7) Slightly toxic (3T3-L1)	[78]
*Artemisia turcomanica* leaf extract	20–60 nm, spherical	n.a.	Cytotoxic (AGS, L929)	[79]
*Juglans regia* walnut green husk	30–50 nm, spherical	*Escherichia coli*, *Pseudomonas aeruginosa*, *Staphylococcus aureus*	Cytotoxic (MCF-7) Non-cytotoxic (L929)	[80]
Black tea (Tetley, England)	30–40 nm, spherical	n.a.	Cytotoxic (A2780) Slightly cytotoxic (HCT116, primary human fibroblast)	[81]
Fenugreek leaf	20–30 nm, spherical	*Escherichia coli*, *Staphylococcus aureus*	Non-cytotoxic (HaCaT)	[82]
*Pimpinella anisum* seed	3–16 nm (average 8.3 nm), spherical	*Staphylococcus pyogenes*, *Acinetobacter baumannii*, *Klebsiella pneumoniae*, *Salmonella typhi*, *Pseudomonas aeruginosa*	Cytotoxic (HT115, hSSCs)	[83]
*Ficus religiosa* leaf extract	3–28 nm (average 21 nm), spherical	*Escherichia coli*, *Pseudomonas fluorescens*, *Bacillus subtilis*, *Salmonella typhi*	Cytotoxic (A549, Hep2, HeLa, COLO205, SH-SY5Y)	[84]
*Alternanthera dentata* leaf	20–90 nm, spherical	*Enterococcus faecalis*, *Escherichia coli*, *Klebsiella pneumoniae*, *Pseudomonas aeruginosa*	n.a.	[85]
*Boerhaavia diffusa* plant	25 nm, spherical	*Aeromonas hydrophila*, *Flavobacterium branchiophilum*, *Pseudomonas fluorescens*	n.a.	[86]
*Tribulus terrestris* dried fruit	16–28 nm, spherical	*Bacillus subtilis*, *Escherichia coli*, *Pseudomonas aeruginosa*, *Staphylococcus aureus*, *Streptococcus pyogenes*	n.a.	[87]
*Cocos nucifera* inflorescence	22 nm, spherical	*Bacillus subtilis*, *Klebsiella pneumoniae*, *Pseudomonas aeruginosa*, *Salmonella paratyphi*	n.a.	[88]
*Abutilon indicum* leaf	7–17 nm, spherical	*Escherichia coli*, *Bacillus subtilis*, *Salmonella typhi*, *Staphylococcus aureus*	n.a.	[89]
*Cymbopogon citratus* fresh leaf	32 nm, spherical	*Escherichia coli*, *Klebsiella pneumoniae*, *Proteus mirabilis*, *Salmonella typhi*, *Staphylococcus aureus, Aspergillus niger*, *Candida albicans*	n.a.	[90]
*Tinospora cordifolia* stem	83 nm, spherical	*Staphylococcus aureus*	n.a.	[91]
*Eucalyptus citriodora* leaf	8–15 nm, spherical	*Acinetobacter baumannii*	n.a.	[92]
*Argemone mexicana* leaf	20 nm, spherical	*Escherichia coli, Pseudomonas aeruginosa*	n.a.	[93]
*Solanum torvum* leaf	14 nm, spherical	*Pseudomonas aeruginosa*, *Staphylococcus aureus, Aspergillus flavus*, *Aspergillus niger*	n.a.	[94]
*Aloe vera* plant	70–192 nm, spherical	*Pseudomonas aeruginosa*, *Streptococcus epidermidis*	Non-cytotoxic (PBMC)	[95]
*Trianthema decandra* root	36–74 nm, spherical	*Bacillus subtilis*, *Enterococcus faecalis*, *Escherichia coli*, *Proteus vulgaris*, *Pseudomonas aeruginosa*, *Staphylococcus aureus*, *Streptococcus faecalis*, *Yersinia enterocolitica, Candida albicans*	n.a.	[96]
*Pongamia pinnata* fresh bark	5–55 nm, spherical	*Bacillus subtilis*, *Klebsiella planticola*, *Klebsiella pneumoniae*, *Staphylococcus aureus*	n.a.	[97]
*Ocimum sanctum* leaf	18 nm, spherical	*Escherichia coli*, *Staphylococcus aureus*	n.a.	[98]
*Catharanthus roseus* leaf	48–67 nm, spherical	*Bacillus cereus*, *Escherichia coli*, *Klebsiella pneumoniae*, *Pseudomonas aeruginosa*, *Staphylococcus aureus*	n.a.	[99]
*Cochlospermum gossypium*	3 nm, spherical	*Escherichia coli*, *Pseudomonas aeruginosa*, *Staphylococcus aureus*	n.a.	[100]
Olive leaf	20–25 nm, spherical	*Escherichia coli*, *Pseudomonas aeruginosa*, *Staphylococcus aureus*	n.a.	[101]
*Withania somnifera* leaf	5–30 nm, spherical	*Aspergillus niger*, *Staphylococcus aureus, Escherichia coli*, *Aspergillus flavus, Candida albicans*	n.a.	[102]
*Datura stramonium* leaf	15–20 nm, spherical	*Escherichia coli*, *Staphylococcus aureus*	n.a.	[103]
*Emblica officinalis* fruit	15 nm, spherical	*Bacillus subtilis*, *Escherichia coli*, *Klebsiella pneumoniae*, *Staphylococcus aureus*	n.a.	[104]
*Crataegus douglasii* fruit	29 nm, spherical	*Escherichia coli*, *Staphylococcus aureus*	n.a.	[105]
*Acalypha indica* leaf	20–30 nm, spherical	*Escherichia coli*, *Vibrio cholerae*	n.a.	[106]
*Solanum indicum* plant	10–50 nm, spherical	*Klebsiella* sp., *Staphylococcus* sp.	Cytotoxic (rat splenocytes)	[107]
*Citrus sinensis* peel	35 nm, 10 nm, spherical	*Escherichia coli*, *Pseudomonas aeruginosa*, *Staphylococcus aureus*	n.a.	[108]
*Hibiscus rosa-sinensis* petal	76 nm, spherical	*Escherichia coli*, *Klebsiella pneumoniae*, *Staphylococcus aureus*, *Vibrio cholerae*	n.a.	[109]
*Daucus carota* fresh extract	20 nm, spherical	*Bacillus cereus*, *Klebsiella pneumoniae*, *Pseudomonas aeruginosa*, *Staphylococcus aureus*	Non-cytotoxic (EAC cells)	[110]
*Melissa officinalis* leaf	12 nm, spherical	*Escherichia coli*, *Staphylococcus aureus*	n.a.	[111]
*Phoenix dactylifera* root hair	15–40 nm, spherical	*Escherichia coli, Candida albicans*	Cytotoxic (MCF-7)	[112]
*Annona muricata* root bark	22 nm	*Bacillus subtilis*, *Staphylococcus aureus, Klebsiella pneumoniae, Escherichia coli, Pseudomonas aeruginosa*	n.a.	[113]
*Terminalia mantaly* fresh leaf, stem bark and root	11–80 nm anisotropic	*Haemophilus influenzae*, *Streptococcus pneumoniae*	n.a.	[114]
*Acacia rigidula* stem and root	22.46 nm spherical	*Escherichia coli,* *Pseudomonas aeruginosa,* *Bacillus subtilis*	Non-toxic in *in vivo* mouse model	[115]
*Lampranthus coccineus* aerial part	10.12–27.89 nm spherical	HAV-10, HSV-1, CoxB4	Non-toxic (HeLa)	[116]
*Malephora lutea* aerial part	8.91–14.48 nm spherical	HAV-10, CoxB4		

See notes on the cell lines at Table 1.

**Table 3 molecules-26-00844-t003:** Characteristics and biological activities of plant-mediated gold nanoparticles.

Organism Used for the Synthesis	Particle Size/Shape	Antimicrobial Effect/Sensitive Species	Effect on Mammalian Cell Lines	Ref.
*Zingiber officinale* rhizome	5–15 nm, spherical	n.a.	No aggregation with human blood cells	[117]
*Mussaenda glabrata* leaf	10–12 nm, spherical	*Bacillus pumilus*, *Staphylococcus aureus*, *Pseudomonas aeruginosa*, *Escherichia coli*, *Aspergillus niger, Penicillium chrysogenum*	n.a.	[72]
*Alpinia nigra* leaf	20–55 nm, spherical and hexagonal	*Bacillus subtilis*, *Escherichia coli*, *Candida albicans*	n.a.	[118]
*Uncaria gambir Roxb.* leaf	11–31 nm, hexagonal, triangular	*Escherichia coli, Staphylococcus aureus*	n.a.	[119]
*Plumbago zeylanica* bark	3–52 nm, spherical	*Escherichia coli*, *Pseudomonas aeruginosa, Bacillus subtilis, Staphylococcus aureus, Candida tropicalis, Aspergillus flavus*	Cytotoxic (Dalton Lymphoma Ascites)	[73]
*Olax scandens* leaf	5–100 nm, mostly spherical	n.a.	Antiproliferative (A549, MCF-7 and COLO 205)	[120]
*Panax ginseng* fresh leaf	10–20 nm, spherical	n.a.	Non-cytotoxic (HaCaT, 3T3-L1)	[74]
*Panax ginseng* berry	5–10 nm, spherical	n.a.	Non-cytotoxic (HDF, B16 murine cell line)	[75]
*Trichosanthes kirilowii* plant	∼50 nm, spherical	n.a.	Cytotoxic (HCT-116)	[121]
*Marsdenia tenacissima* plant	∼50 nm, spherical (anisotropic shapes)	n.a.	Cytotoxic (A549)	[122]
*Nerium oleander* stem bark	10–100 nm, mostly spherical	n.a.	Cytotoxic (MCF-7) Non-cytotoxic (Human lymphocytes)	[123]
*Tribulus terrestris* dried fruit	7 mm and 55 mm, mostly spherical	multi-drug resistant strains of *Helicobacter pylori*	GNP7 cytotoxic at 200 μg/mL (AGS) GNP55 less cytotoxic (AGS) GNP7 and GNP55 non-cytotoxic at the MIC of *H. pylori* (AGS)	[124]
*Peltophorum pterocarpum* leaf	∼55 nm, primarily spherical	n.a.	Non-cytotoxic (HUVEC, ECV-304)	[125]
*Dendropanax morbifera* leaf	10–20 nm; polygonal and hexagonal	n.a.	Non-cytotoxic (A549, HaCaT)	[77]
*Anemarrhena asphodeloides* rhizome	10 nm, crystalline face-centered cubic	n.a.	Non-cytotoxic, toxic only in high concentration (100 ug/mL) (A549, HT29, MCF-7, 3T3-L1)	[78]
Potato starch powder	20–30 nm, quasi-spherical	MDR *E. coli*, MRSA	Non-toxic (human dermal fibroblasts, melanoma cells)	[126]
	Ag-Au composite: 9–10 nm, quasi-spherical		Cytotoxic (human dermal fibroblasts) Enhanced cytotoxic (melanoma cells)	
*Trapa natans* peel extract	Ag-Au composite: 26–90 nm, hexagonal, triangular, spherical	n.a.	Cytotoxic (HeLa, MDA-MB-231, HCT116 cancer cells)	[127]
*Trianthema decandra* root	33–65 nm, spherical, triangular, hexagonal and cubical	*Bacillus subtilis*, *Enterococcus faecalis*, *Escherichia coli*, *Proteus vulgaris*, *Pseudomonas aeruginosa*, *Staphylococcus aureus*, *Streptococcus faecalis*, *Yersinia enterocolitica, Candida albicans*	n.a.	[96]
*Ananas comosus* fruit	16 nm, anisotropic	*Streptobacillus* sp., *Escherichia coli*	n.a.	[128]
*Annona muricata* leaf	25.5 nm, spherical	*Staphylococcus aureus*, *Enterococcus faecalis*, *Klebsiella pneumoniae*, *Clostridium sporogenes, Aspergillus flavus, Candida albicans, Fusarium oxysporum, Penicillium camemberti*	n.a.	[129]
*Allium cepa*	11 nm, spherical	measles virus	n.a.	[130]
*Azima tetracantha* leaf	80 nm, spherical	*Aeromonas liquefaciens, Enterococcus faecalis, Micrococcus luteus, Salmonella typhimurium*, *Candida albicans, Cryptococcus* sp., *Microsporum canis, Trichophyton rubrum*	n.a.	[131]
*Caulerpa racemosa* green seaweed	13.7–85.4 nm, spherical to oval	*Aeromonas veronii*, *Streptococcus agalactiae*	Cytotoxic (HT-29)	[132]
*Nepenthes khasiana* leaf	50–80 nm, spherical	*Escherichia coli*, *Bacillus* sp., *Aspergillus niger*, *Candida albicans*	n.a.	[133]
*Salix alba*	63 nm, spherical	*Staphylococcus aureus*, *Alternaria solani, Aspergillus niger, Aspergillus flavus*	n.a.	[134]
Curcumin	11.95 nm, lattice	respiratory syncytial virus (RSV)	n.a.	[135]
*Abelmoschus esculentus* pulp	14 nm, spherical	*Bacillus subtilis, Bacillus cereus, Pseudomonas aeruginosa, Micrococcus luteus, Escherichia coli*	Cytotoxic (Jurkat cells)	[136]
*Spirulina platensis* green alga	5 nm, spherical	*Staphylococcus aureus, Bacillus subtilis*		[137]
*Halymenia dilatata* red alga	16 nm, spherical, triangular	*Aeromonas hydrophila*	Cytotoxic (HT-29)	[138]
*Acanthophora spicifera* seaweed	˃20 nm, spherical	*Vibrio harveyi, Staphylococcus aureus*	Cytotoxic (HT-29)	[139]
*Gelidium pusillum* seaweed	∼55 nm, spherical	n.a.	Cytotoxic (MDA-MB-231) Non-cytotoxic (HEK-293)	[140]

See notes on the cell lines at Table 1.

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
