# Peer review of "Green Silver and Gold Nanoparticles: Biological Synthesis Approaches and Potentials for Biomedical Applications"

_molecules, 2021, doi:10.3390/molecules26040844_

Round 1

Reviewer 1 Report

The authors have reviewed the recent progress on the green synthesis of gold and silver nanoparticles and their biomedical applications including antibacterial, antifungal and anticancer properties. Although a large number of review articles have been published recently on the green synthesis of metallic nanoparticles that emphasize the challenges and their applications in various fields. Nevertheless, more advanced level knowledge can facilitate the readers with existing challenges and their solutions as well as how these approaches can be modified to achieve large scale, sustainable synthesis of metallic nanoparticles and improve their performance for different applications. The present review article mainly summarizes the recent work but failed to describe the challenges associated with green synthesis systematically, thus in my opinion can only be accepted for publication in Molecules if it can be revised carefully considering the current requirement of review articles. Additional questions are listed below.

  1. The article discusses the biological synthesis of gold and silver nanoparticles, however, recent literature about the application of other bio reducing agents for the green synthesis of these materials must be included, such as ascorbic acid, citrate, amino acid and glucose etc.
  2. The major part of the review article comprises of bio-synthesis of gold and silver NPs but the authors only described their general methods of synthesis. The authors should explain the challenges and specified requirements to achieve these materials on large scale with high quality using these reducing agents.
  3. The authors just list out few biomedical applications based on gold and silver NPs, however these materials have several other biomedical applications including biosensors, imaging, diagnostics etc. Therefore, authors should include another section to highlights recent advancement in these applications.
  4. The authors are suggested to discuss these articles and cite them in their review article: Dalton Transactions 47.35 (2018): 11988-12010; Sustainability 12.4 (2020): 1484; Biomolecules 10.3 (2020): 452; Nanomaterials 10.9 (2020): 1763. International Journal of Nanomedicine 15 (2020): 275. Nanomaterials 10.9 (2020): 1885.
  5. There are a lot of grammatical and spelling mistakes in the manuscripts, so authors must revise the manuscript carefully.
  6. The review contains only one figure that too the quality is not good. The authors must include few figures and cite.

Author Response

Response to the Reviewer

We are grateful to the Reviewer for evaluating our manuscript. We highly appreciate the feedback and the detailed and valuable comments of the Referee. The suggestions were straightforward and we addressed them in the revised manuscript. We feel that the overall value of the manuscript has been substantially improved by incorporating these suggestions and we hope that the Reviewer will be satisfied with the revised manuscript. Please find below our detailed response to the comments and questions.

The authors have reviewed the recent progress on the green synthesis of gold and silver nanoparticles and their biomedical applications including antibacterial, antifungal and anticancer properties. Although a large number of review articles have been published recently on the green synthesis of metallic nanoparticles that emphasize the challenges and their applications in various fields. Nevertheless, more advanced level knowledge can facilitate the readers with existing challenges and their solutions as well as how these approaches can be modified to achieve large scale, sustainable synthesis of metallic nanoparticles and improve their performance for different applications. The present review article mainly summarizes the recent work but failed to describe the challenges associated with green synthesis systematically, thus in my opinion can only be accepted for publication in Molecules if it can be revised carefully considering the current requirement of review articles. Additional questions are listed below.

We thank the Reviewer for pointing out that the manuscript would benefit from adding information about the challenges associated with green synthesis systematically. Therefore, we revised the nanoparticle synthesis part of the manuscript and according to the suggestion of the Reviewer, we complemented the manuscript with a chapter dealing with the available data as well as with our view on the existing challenges of green nanoparticle synthesis. We hope that the Reviewer agrees on the problems, challenges and critical factors we have highlighted in this additional section. Obviously, these factors need to be controlled and considered at the design of such materials in order to minimize their impact upon synthesis, as much as it is possible.

  1. The article discusses the biological synthesis of gold and silver nanoparticles, however, recent literature about the application of other bio reducing agents for the green synthesis of these materials must be included, such as ascorbic acid, citrate, amino acid and glucose etc.

“Green synthesis” of nanomaterials can be defined in various ways. The concepts of such synthetic approaches can vary depending on how broadly or strictly the idea of “green synthesis” is considered.

Our view on “green synthesis” comprises the production of nanoparticles (or nanomaterials in general) by utilizing either living entities (for example a certain plant), or parts of living entities (such as plant leaf, stem, fruit and extracts obtained from these) upon the synthesis. According to this, utilization of purified single organic molecules (such as citrate or ascorbic acid, etc.) during nanomaterial synthesis does not strictly belong to the conventional concept of “green synthesis”.

We are certainly aware of the fact that citrate, amino acids, glucose and ascorbic acid are often applied in nanoparticle synthesis, however, such “bio reducing agents” are usually obtained by conventional chemical production, despite the possibility of their industrial synthesis without hazardous chemicals.  Although their utilization for nanoparticle synthesis requires milder conditions compared to traditional reductants (like NaBH4), in our view such an approach is not “green synthesis” in the conventional sense. Based on this idea and these concepts, we did not include classic chemical nanoparticle synthesis methods using industrially produced, purified, simple unicomponent organic molecules such as citrate as reducing agents into the present review. Discussing all these possible purified “bio reducing agents” and their involvement in nanoparticle production would definitely exceed the acceptable dimension of the manuscript.

We see the point the Reviewer has raised, however, our concept of “green synthesis” is probably a more stringent approach. We are cordially asking the Reviewer to accept this notion.

  1. The major part of the review article comprises of bio-synthesis of gold and silver NPs but the authors only described their general methods of synthesis. The authors should explain the challenges and specified requirements to achieve these materials on large scale with high quality using these reducing agents.

We thank the Reviewer for this suggestion. According to her/his suggestion, we have complemented the manuscript with a chapter about the challenges associated with green synthesis and with information about the specific requirements these synthesis methods need for small-scale and also for larger-scale nanoparticle production. We hope that the Reviewer agrees on the problems, challenges and critical factors we have emphasized here in this additional section of the revised manuscript.

  1. The authors just list out few biomedical applications based on gold and silver NPs, however these materials have several other biomedical applications including biosensors, imaging, diagnostics etc. Therefore, authors should include another section to highlights recent advancement in these applications.

We completely agree with the Reviewer that the application of gold and silver nanoparticles as biosensors, or in medical diagnostics and imaging are extremely important biomedical functions of such nanomaterials. When we first constructed the skeleton of the manuscript, in fact, we planned to include a chapter about these utilization possibilities as well. However, this chapter did not materialize in the submitted version of the manuscript owing to the intention to restrict the size of the paper to an acceptable length. Nevertheless, as the Reviewer recommended to include this information anyway, we complied to her/his request, thus these biomedical applications are now summarized in a separate chapter of the revised manuscript.

  1. The authors are suggested to discuss these articles and cite them in their review article: Dalton Transactions 47.35 (2018): 11988-12010; Sustainability 12.4 (2020): 1484; Biomolecules 10.3 (2020): 452; Nanomaterials 10.9 (2020): 1763. International Journal of Nanomedicine 15 (2020): 275. Nanomaterials 10.9 (2020): 1885.

The publications suggested by the Reviewer have been included into the revised version of the manuscript and these papers are cited in the manuscript accordingly.

  1. There are a lot of grammatical and spelling mistakes in the manuscripts, so authors must revise the manuscript carefully.

The manuscript has been screened for spelling mistakes, grammatical errors and spacing mistakes and has been corrected accordingly.

  1. The review contains only one figure that too the quality is not good. The authors must include few figures and cite.

The quality of the figure in the submitted manuscript adhered to the requirement described in the instructions for authors. The problem with the resolution may have appeared due to the fact that the image had to be placed within the template of the manuscript. To avoid any problems regarding the image, or its resolution, we will upload the figure separately from the manuscript onto the manuscript submission site.

Furthermore, we have prepared two additional figures explaining the nanoparticle “green synthesis” and the numerous biomedical applications in further detail. We hope that the 3 figures and the 3 tables of the revised manuscript can highlight all relevant aspects of “green” nanoparticle production and utilization.

Reviewer 2 Report

This review paper well summarized the biosynthesis and biomedical uses of silver and gold nanoparticles. It should have broad interest to the researchers working with noble metal nanoparticles for biomedical applications. Generally, the paper is well organized and most of the relevant topics are fully covered. The manuscript is well written and the references are properly cited. It is will be great if the authors can make it more concise, as the paper is too long and most of the details of the reference paper are not necessary. I am very pleased to recommend its publication in the journal of molecules if the following questions are appropriately addressed.

  1. In the biological synthesis section, the authors only focus on the biosynthesis of gold and silver nanoparticles, and the purification strategies are not fully covered. It will benefit the researchers in this field if this part of the information can be provided in the manuscript.
  2. Only the biomedical uses related to their biological activity are discussed, but the applications such as drug/gene delivery, disease diagnosis, photothermal therapy, and so on, are missing. The authors can either revise the title of the manuscript or add the other applications to the manuscript.
  3. Most of the biomedical uses are regarding in vitro studies, it will be great if the authors can introduce some in vivo findings if there are any.

Author Response

Response to the Reviewer

We are very grateful to the Reviewer for the positive evaluation of our manuscript. We honestly appreciate the feedback, the suggestions and the questions, which helped to improve considerably the quality of the manuscript. All the issues raised by the Reviewer are addressed in the revised manuscript and we hope that the Reviewer will be satisfied with the modifications. Please find below our detailed response to the comments and questions.

This review paper well summarized the biosynthesis and biomedical uses of silver and gold nanoparticles. It should have broad interest to the researchers working with noble metal nanoparticles for biomedical applications. Generally, the paper is well organized and most of the relevant topics are fully covered. The manuscript is well written and the references are properly cited. It is will be great if the authors can make it more concise, as the paper is too long and most of the details of the reference paper are not necessary. I am very pleased to recommend its publication in the journal of molecules if the following questions are appropriately addressed.

According to the suggestion of the Reviewer, and considering the chapters the other Reviewers required, the paper has been substantially modified. Some parts have been compressed, thus were rendered more concise, however, other parts have been augmented based on the suggestions of the Referees. We have done our best to reduce the length of some chapters and also to eliminate the details which are not essential for the present review. Despite these efforts, due to the added information, the manuscript has not been shortened significantly. 

  1. In the biological synthesis section, the authors only focus on the biosynthesis of gold and silver nanoparticles, and the purification strategies are not fully covered. It will benefit the researchers in this field if this part of the information can be provided in the manuscript.

We thank the Reviewer for calling our attention to the point that in the synthesis chapter of the manuscript the purification strategies are not fully covered and further information on this issue would assist the projects of other researchers. Therefore, we revised the nanoparticle synthesis part of the manuscript and according to the suggestion of the Reviewers, we complemented this chapter with the available data on the purification strategies and on the challenges related to this step of green nanoparticle synthesis.

2. Only the biomedical uses related to their biological activity are discussed, but the applications such as drug/gene delivery, disease diagnosis, photothermal therapy, and so on, are missing. The authors can either revise the title of the manuscript or add the other applications to the manuscript.

We agree with the Reviewer that besides the biological activity (such as antimicrobial and anticancer features) of gold and silver nanoparticles their application potential as drug and gene delivery platforms, as well as in medical diagnostics and in photothermal therapy are extremely important. Therefore, we prepared a new chapter dealing with these utilization possibilities of noble metal nanoparticles, which are now summarized in the revised manuscript.

3. Most of the biomedical uses are regarding in vitro studies, it will be great if the authors can introduce some in vivo findings if there are any.

We wanted to summarize the findings of all in vivo research projects, where green synthesized gold or silver nanoparticles were administered to living organisms systemically. Thus, we have gathered the available data on in vivo applications. However, we have to note that most toxicology studies and projects dealing with the biological efficiency and with the triggered molecular mechanisms of these types of nanoparticles are performed on in vitro cell cultures, rather than on in vivo systems. This data is seriously lacking in the scientific literature. We believe that the main reason for this lack of information is based on the simpler and economical approach of using cell cultures for screening the biological performance of “green” nanoparticles over animal experiments, as well as owing to the guiding principles for human and animal research for drug development. These principles and guidelines state that animal experiments must be replaced wherever possible by other methods such as mathematical modeling or by in vitro biological systems to avoid or reduce the number of animals sacrificed for research purposes. Nevertheless, some in vivo studies have been completed, and we made a great effort to find them all. These studies are discussed and cited in the revised manuscript.

Reviewer 3 Report

this review is well written and organized however, several questions that did not mention in this paper.

  1. what is the size ranges of Ag and Au NPs by using microorganism and plant extract to facility to synthesize?
  2. Also does it has an average sizes??
  3. compare to chemical synthesis using chemicals and microorganism and plant extract, which one yield more, and which one has a better size and quality for commercial usage?
  4. table 1 is too long ,put into supplements.
  5. For antimicrobial activities, Au Ag NPs from chemicals and microorganism and plant extract, which one is better?
  6. what is benefit for Ag and Au NPs by using microorganism and plant extract to facility to synthesize, economic , environmental friendly? do you have this data???
  7. need to address these questions in the text too.
  8.  

Author Response

Response to the Reviewer

We are very grateful to the Reviewer for her/his positive opinion on our manuscript. We thank the questions raised by the Reviewer and we appreciate the opportunity to provide our view on these issues. We incorporated these suggestions into the revised manuscript. Please find below our detailed response to the comments and questions.

this review is well written and organized however, several questions that did not mention in this paper.

  1. what is the size ranges of Ag and Au NPs by using microorganism and plant extract to facility to synthesize?

The size range of gold or silver nanoparticles produced by microorganisms and plant extracts varies enormously depending on the biological materials used for the synthesis. Generally, nanoparticle size is under the control of the molecular composition of the green material, the temperature set for the synthesis as well as the velocity of stirring the colloid upon nanoparticle formation. The concentration and the chemical nature of the molecules/ions within the green material are the most important determining factors. If this solution contains compounds that have a strong reducing potential, that means the gold or silver metal ions would be reduced fairly quickly, yielding a colloid with a high number of rather small particles. If the combined reducing potential of the bioactive molecules in the green solution is weak, the nanoparticle formation often leads to larger-sized particles or the reduction of ions is not always complete. Furthermore, the stability and the aggregation propensity of the obtained nanoparticles can also vary depending on the molecules of the green solution. As the exact composition (qualitative and quantitative analysis) of the green material, let it be a plant extract or a microbial lysate, is not known precisely, the size range of the obtained nanoparticles cannot be designed precisely prior to the actual synthetic process. The more complex the composition of the green solution is, the more problematic the estimation of the expected nanoparticle size range becomes. Nevertheless, a precise description of the materials and conditions used (pH, time, temperature, stirring velocity) for the synthesis allows other researchers to regulate and fine-tune the nanoparticle size range upon production (Khan M et al https://doi.org/10.1039/c8dt01152d; Lee KX et al https://doi.org/10.2147/IJN.S233789).

Here we list some examples of the different factors influencing nanoparticle production especially considering nanoparticle size. According to Gröning et al (Die Pharmazie, 2001, 56(10), 790-792.), in the presence of tea, particles in smaller numbers were formed, while coffee catalyzed the production of a larger amount of nanoparticles despite the same input quantity. More importantly, it was also observed that by increasing the concentration of the green tea extracts, the size of the nanoparticles decreased while their number increased. The size and shape selectivity of gold and silver nanoparticles has been achieved using Aloe vera leaf extract as a reducing agent (Chandran et al https://doi.org/10.1021/bp0501423). By changing the initial salt and extract volume and concentration used for the synthesis of nanoparticles during the reaction had a marked impact on the production of characteristic nanoparticles. The effect of synthesis time, temperature, pH, stirring time, and extract concentration and volume, have also been investigated. For instance, it was demonstrated that the control over the size of metal nanoparticles is time-dependent. Generally, the longer the reaction time was, the larger the size of the nanoparticles were (Elemike et al https://doi.org/10.1007/s12034-017-1362-8). In addition, elevating the stirring time increased the mean particle size (Balavandy et al https://doi.org/10.1186/1752-153X-8-11). Regarding the temperature, it was proven that the mean particle size and the number of the obtained nanoparticles decreased with increasing temperature. Moreover, as the binding of metal ions to the biological component is pH-dependent, thereby diverse such as tetrahedral, hexagonal platelets, rod-shaped, irregular-shaped particles can form, and smaller nanoparticles can be achieved at higher pH (Armendariz et al https://doi.org/10.1007/s11051-004-0741-4).

2. Also does it has an average sizes??

Generally, silver and gold nanoparticles obtained by green approaches are often between 2-100 nm diameter in size, according to the accumulated and cited publications, but rarely, larger particles are also produced having an average diameter of 150 or even 200 nm (see Tables 1,2 and 3). Although these fairly large nanoparticles are usually ineffective, green nanoparticles within the commonly obtained size range are suitable for biomedical applications.

3. compare to chemical synthesis using chemicals and microorganism and plant extract, which one yield more, and which one has a better size and quality for commercial usage?

According to our knowledge and based on a thorough survey on the websites of some companies, the commercially available nanoparticle samples are all prepared using classic chemical reduction methods. Indeed, the yield of such a classic chemical nanoparticle synthesis can be calculated typically by considering the chemical reactions, equations, and the amount of metal ions applied upon the synthesis, and of course, AgNP or AuNP colloids are available in various size ranges with different capping and stabilizing materials according to the requirements of the customer. Although the companies generally do not publish every detail of the metal colloid they are selling, some data on the purity or quality of these products are available. Obviously, green synthesized noble metal nanoparticles are more complicated in this regard. It is more difficult to describe certain parameters and the precise quality and quantity of the components within such a nanoparticle colloid. However, the yield of a green nanoparticle manufacturing method can potentially be just as high as of a chemical method and the biological efficiency can also match or even exceed that of a chemically obtained colloid. Considering the size of nanoparticles, the ideal size depends enormously on the actual utilization purpose, especially when the particles are destined for biomedical applications. High toxicity can be achieved by applying small-sized nanoparticles, which can definitely be produced both by “green” as well as by conventional chemical manufacturing. Under specific circumstances, larger nanoparticles are needed, which are also easily realized by “green” and also by chemical approaches. Therefore, it is impossible to draw a clear conclusion or reach a definite decision on which nanoparticle is “better”.

4. table 1 is too long ,put into supplements.

We thank the Reviewer for the suggestion. In fact, the tables in the manuscript are quite long. To consider the suggestion of the Reviewer but at the same time to maintain the option to check on any nanoparticle (its synthesis and biological performance) upon reading the appropriate chapter of the manuscript, we dissected Table 2. of the original manuscript and generated 2 smaller tables, that were incorporated into the body of the manuscript. Furthermore, all 3 tables were simplified considerably, as every non-essential information has been deleted from the tables. We considered the Reviewer’s option to place the original tables into the Supplementary. However, our strong impression is that the information positioned in the supplementary material is often neglected and overseen, and it fails to fulfill its role as an important component of the manuscript. We hope that the Reviewer can accept this solution about the fate of the tables.

5. For antimicrobial activities, Au Ag NPs from chemicals and microorganism and plant extract, which one is better?

We thank the Reviewer for raising this question because comparing the biological activity of the newly synthesized nanoparticles to a standard should be an essential part of their biomedical characterization regimen. However, there is only a limited amount of data available in the literature about similar comparison approaches. Ferreyra Maillard et al (https://doi.org/10.1016/j.colsurfb.2018.07.044) studied the antibacterial effect of green and chemically synthesized AgNPs having the same morphology. They demonstrated that green synthesized AgNPs exhibited stronger antibacterial activity than chemically synthesized counterparts, however, the size of the latter particles was smaller. Garibo et al (https://doi.org/10.1038/s41598-020-69606-7) also compared the antimicrobial activity of chemically and Lysiloma acapulcensis-mediated green synthesized AgNPs. They found that green synthesized AgNPs exerted higher antimicrobial activity than the chemically produced counterparts upon applying them in the same concentration. However, in this publication, we could not find data about the size and the shape of the chemically synthesized AgNPs, despite the fact that these factors would significantly influence the effectiveness of the NPs. In a previous study from us and our collaborators (Bélteky et al https://doi.org/10.2147/IJN.S185965), we described the behavior of AgNPs produced by citrate and green tea extract. Both the size and the shape of the as-prepared particles were similar. By applying these particles at the same concentration in antimicrobial tests, the green tea‑synthesized AgNPs proved to be more effective antimicrobials than the chemically synthesized ones. These results demonstrated that green synthesized AgNPs have the potential to be biologically more active than chemically synthesized counterparts, even if the physical properties of the two different AgNP colloids were very similar.

6. what is benefit for Ag and Au NPs by using microorganism and plant extract to facility to synthesize, economic , environmental friendly? do you have this data???

This issue is indeed very important, since apart from the basic science aspect, namely to achieve silver and gold nanoparticle synthesis in alternative ways other than the classic chemical synthesis options, the “green” approach of nanoparticle manufacturing has environmental as well as economic effects. From the environmental aspect, “green” synthesis is highly advantageous, since the entities that can be utilized for this purpose are endless, and the application of harsh conditions, dangerous chemicals, and the disposal of toxic wastes can be avoided. The economical aspect is certainly an equally prominent element to consider. These “green” materials are often cheap, they are available in high quantities and the production of “green solution” for nanoparticle synthesis using these entities is generally very cost-effective. To estimate the precise economic benefits of utilizing “green” materials especially in a higher-scale/industrial-scale production of silver and gold nanoparticles versus classic chemical approaches needs to be analyzed by specialists and only in view of these numbers can clear conclusions to be drawn regarding the financial and economic aspects of “green” approaches.

7. need to address these questions in the text too.

We thank the Reviewer again for pointing out these issues. We have addressed them in the revised version of the manuscript.

Round 2

Reviewer 1 Report

The authors have made all the requested changes in the manuscript now it is acceptable for publication.